# Deep Research Agents:
# A Systematic Examination And Roadmap

## Abstract

The rapid progress of Large Language Models (LLMs) has given rise to a new category of autonomous AI systems, referred to as Deep Research (DR) agents. These agents are designed to tackle complex, multi-turn informational research tasks by leveraging a combination of dynamic reasoning, adaptive long-horizon planning, multi-hop information retrieval, iterative tool use, and the generation of structured analytical reports. In this paper, we conduct a detailed analysis of the foundational technologies and architectural components that constitute Deep Research agents. We begin by reviewing information acquisition strategies, contrasting API-based retrieval methods with browser-based exploration. We then examine modular tool-use frameworks, including code execution, multimodal input processing, and the integration of Model Context Protocols (MCPs) to support extensibility and ecosystem development. To systematise existing approaches, we propose a taxonomy that differentiates between static and dynamic workflows, and we classify agent architectures based on planning strategies and agent composition, including single-agent and multi-agent configurations. We also provide a critical evaluation of current benchmarks, highlighting key limitations such as restricted access to external knowledge, sequential execution inefficiencies, and misalignment between evaluation metrics and the practical objectives of DR agents. Finally, we outline open challenges and promising directions for future research.

## 1 Introduction

Recent advances in large language models (LLMs) have led to the rapid emergence of AI agents capable of autonomous research. Early models such as GPT-3 (Brown et al., 2020) primarily addressed isolated tasks, including question answering and machine translation. Subsequently, integration with external tools enabled models such as WebGPT (Nakano et al., 2021) to navigate the web and synthesise information from diverse sources autonomously. Most recently, a new class of autonomous systems, termed Deep Research (DR) agents, has emerged; publicly announced commercial examples include OpenAI DR (OpenAI, 2025b), Gemini DR (Google Team, 2025), Grok DeepSearch (xAI Team, 2025), and Perplexity DR (Perplexity Team, 2025). These deep research agents extend LLMs by incorporating advanced reasoning, dynamic task planning, and adaptive interaction with web resources and analytical tools. Figure 5 provides a chronological overview of DR agents evolution.

Formally, we define **"Deep Research Agents"** as:

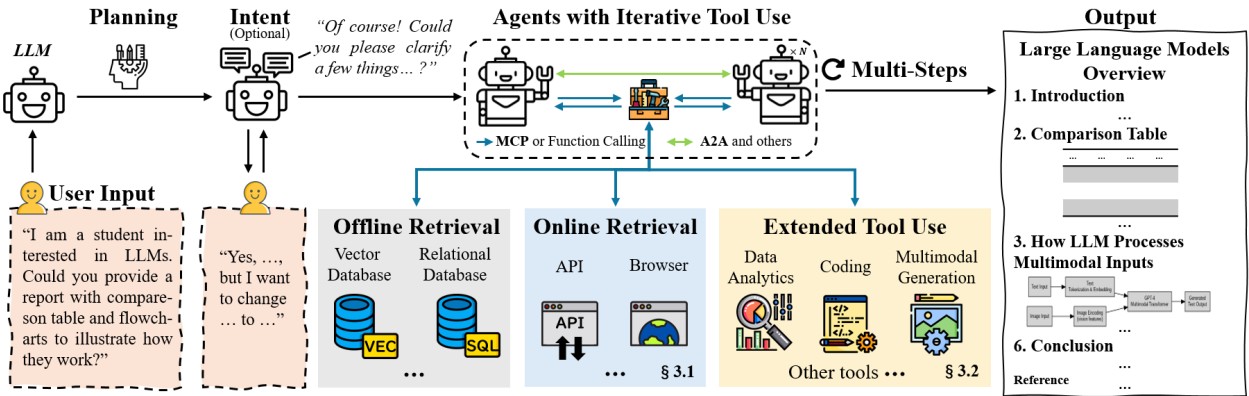

Figure 1: Overview of the DR agent architecture. The workflow is structured into three phases: (1) Input Processing, which begins with planning and an optional intent clarification phase to refine user queries; (2) Core Execution, involving agents with iterative tool use and multi-step reasoning loops to leverage offline retrieval (vector/relational databases), online retrieval (APIs/browsers, see 3.1), and extended tools (coding/multimodal generation, see 3.2); and (3) Output Generation, which produces a structured, comprehensive report containing text, tables, and diagrams.

*AI agents powered by LLMs, integrating dynamic reasoning, adaptive planning, and iterative tool use to acquire, aggregate, and analyse external information, culminating in comprehensive outputs for accomplishing open-ended informational research tasks.*

Specifically, DR agents leverage LLMs as their cognitive core, retrieving external knowledge in real-time through web browsers and structured APIs, and dynamically invoking analytical tools via customised toolkits or standardised interfaces such as the Model Context Protocol (MCP). This architecture enables DR agents to autonomously manage complex, end-to-end research workflows by seamlessly integrating reasoning processes with multimodal resources.

Compared with traditional Retrieval-Augmented Generation (RAG) methods (Singh et al., 2025), which primarily enhance factual accuracy but lack sustained reasoning capabilities (Chen et al., 2025c), and conventional Tool Use (TU) systems (Qu et al., 2025) that heavily depend on pre-defined workflows (Wang et al., 2025), DR agents incorporate autonomous operation, extended reasoning chains, dynamic task planning, and real-time interaction with external sources. These capabilities make DR agents a plausible fit for complex, evolving, and knowledge-intensive research scenarios. A representative example of such a DR agent architecture is illustrated in Figure 1, which demonstrates the complete workflow from user input through optional planning and intent clarification, to iterative tool utilization encompassing offline retrieval (vector and relational databases), online retrieval (APIs and browsers), and extended capabilities including data analytics, coding (etc.), and multimodal generation, ultimately producing comprehensive structured report.

**Contribution.** This survey systematically reviews recent advancements in DR agents, providing a comprehensive analysis of core technologies, methodologies, optimisation pipelines, and representative implementations. Specifically, the contributions of this survey include:

- A thorough analysis of representative DR systems, explicitly examining their system architectures, retrieval mechanisms, tool invocation methods, and performance characteristics, alongside optimisation and tuning paradigms.

- A unified classification framework (Figure 4) that systematically categorises DR systems based on workflow characteristics (static versus dynamic), planning strategies, and agent-based architectures (single-agent versus multi-agent), bridging diverse technical methodologies and current industrial solutions.

- A systematic review and categorisation of existing benchmarks utilised to evaluate DR systems, highlighting how these benchmarks assess critical capabilities, such as retrieval accuracy, reasoning depth, and adaptive tool invocation proficiency.

- A systematic analysis of critical open challenges and research directions, focusing on expanding retrieval scope beyond traditional methods, enabling asynchronous parallel execution, developing comprehensive multi-modal benchmarks, and optimising multi-agent architectures for enhanced robustness and efficiency.

**Survey Organization.** This survey methodically explores recent advancements in DR agents, organised as follows: Section 2 provides foundational concepts, examining recent progress in reasoning, retrieval-augmented generation, and agent communication protocols. Section 3 comprehensively analyses key DR agent components, including search engine integration (Section 3.1), tool invocation strategies (Section 3.2), architectural workflows (Section 3.3), and optimisation methodologies (Section 3.4). Section 4 reviews major industrial applications and practical implementations of DR agents by leading organisations. Section 5 surveys benchmarks used for evaluating DR systems, categorising them into question-answering and task execution scenarios. Section 6 highlights critical challenges and outlines promising directions for future research, focusing on enhancing information acquisition, asynchronous parallel execution, benchmark alignment, and optimising multi-agent architectures. Finally, Section 7 concludes with a summary and provides insights into the broader implications and opportunities within DR agent research.

## 2 Background and Preliminaries

### 2.1 Advances in Reasoning and Tool Integration

Recent advancements in large reasoning models (LRMs) have greatly enhanced the ability of language models to tackle complex and abstract tasks. These models have shown significant improvements in tasks such as arithmetic, common-sense reasoning, and symbolic problem-solving, largely due to innovations in model architectures and training techniques. One such advancement is Chain-of-Thought (CoT) prompting, introduced by Wei et al. (Wei et al., 2023), which explicitly guides models to articulate intermediate logical steps, decomposing complex problems into simpler, sequential stages. This has led to measurable improvements in LLM performance on reasoning benchmarks such as GSM8K and MATH (Cobbe et al., 2021; Hendrycks et al., 2021). Building upon CoT, subsequent research has introduced methods to further enhance LLM reasoning, particularly in handling lengthy textual contexts. Approaches such as positional interpolation and sparse attention mechanisms (Bai et al., 2024a; Wang et al., 2024b) have been proposed to extend the effective context window. Furthermore, specialised benchmarks like LongBench (Bai et al., 2024b) and LongFinanceQA (Lin et al., 2025) have been developed to rigorously evaluate and improve the performance of these models in extended-context reasoning.

To address reasoning tasks that require real-time or specialised external knowledge, frameworks like Tool-former (Schick et al., 2023) and MultiTool-CoT (Inaba et al., 2023) have been proposed, enabling LLMs to autonomously incorporate external computational resources and APIs directly within reasoning workflows. These approaches effectively enhance performance in tasks dependent on precise numerical calculations and dynamic information retrieval. Maintaining reasoning coherence across multiple conversational turns also poses distinct challenges. Techniques such as Dialogue CoT (Chae et al., 2023) and Structured CoT (SCoT) (Sultan et al., 2024) explicitly integrate dialogue states and conversational context within reasoning chains, and have been reported to improve coherence and context-awareness in multi-turn settings, supporting iterative interactions and clarification of complex user queries. However, despite substantial improvements, existing reasoning frameworks still encounter critical issues, including hallucinations, static or outdated internal knowledge, and insufficient responsiveness to rapidly changing information needs. These limitations highlight the necessity of integrating external information sources, real-time retrieval mechanisms, and adaptive reasoning strategies, which are core motivations driving recent advances toward more comprehensive and robust reasoning frameworks suitable for DR Agent applications.

## 2.2 Advances in Retrieval-Augmented Generation and Agentic Retrieval

Retrieval-augmented Generation (RAG), leveraging external knowledge bases (e.g., webs, APIs), has emerged as an effective strategy to mitigate hallucination problems and enhance the accuracy of web information search (Fan et al., 2024; Gao et al., 2023; Singh et al., 2025). Early RAG architectures typically involved a static pipeline, where retrievers fetched relevant documents from external sources such as Wikipedia or search engines, and generators (e.g., LLMs) produced answers based solely on these retrieved passages. However, static approaches were limited in handling complex or multi-step queries, motivating recent advances toward iterative and interactive retrieval mechanisms to generate richer and more relevant responses, including FLARE (Zhang et al., 2024), Self-RAG (Asai et al., 2023), IAG (Zhang et al., 2023), and ToC (Kim et al., 2023). In addition, studies (Izacard et al., 2023; Lin et al., 2023) expanded retrieval sources from structured databases (e.g., Wikipedia) to large-scale, diverse web corpora such as the Common Crawl dump preprocessed via the CCNet pipeline (Fu et al., 2022). Further improvements of RAG include hybrid approaches that combine internal LLM knowledge and external retrievals for better accuracy and coherence (Aliannejadi et al., 2024). Recently, Huang et al. (Huang et al., 2025) proposed RAG-RL, introducing reinforcement learning and curriculum learning techniques, enabling reasoning language models (RLMs) to more effectively identify and utilise relevant contexts. While RAG frameworks may access external sources ranging from pre-indexed corpora to live web APIs, the majority of existing implementations remain anchored to static or periodically refreshed document collections, with truly real-time retrieval capabilities emerging only in recent agentic extensions.

Despite these advancements in retrieval methods and reasoning-enhanced models, RAG approaches still face limitations in effectively managing complex reasoning workflows and dynamically adapting to varied task requirements. To address these challenges, recent research extends RAG into an agentic paradigm, integrating additional reasoning and decision-making layers atop conventional RAG pipelines (Singh et al., 2025). Agentic RAG approaches leverage iterative retrieval, adaptive querying, and dynamic workflow adjustments, aiming to enhance multi-step reasoning capabilities. For example, RL-based query refinement techniques (e.g., Hsu et al. (Hsu et al., 2024)) improve retrieval for complex queries, while graph-based retrieval (e.g., GeAR (Shen et al., 2024)) further enhances the processing of multi-hop queries. Despite these advancements, agentic RAG still faces critical challenges, including balancing computational overhead from dynamic reasoning processes (Singh et al., 2025), aligning agent behaviours with user intentions (Zerhoudi & Granitzer, 2024), and ensuring interpretability in adaptive workflows (Hsu et al., 2024; Singh et al., 2025). Moreover, even advanced agentic

RAG approaches remain constrained by their reliance on pre-existing or periodically updated corpora, limiting their ability to handle real-time, rapidly changing, or long-tail information needs effectively. Addressing this challenge requires integrating external APIs and web browsing capabilities into RAG architectures, motivating recent DR methods aimed at further enhancing retrieval comprehensiveness and adaptability.

### 2.3  Model Context Protocol and Agent-to-Agent Policy

Model Context Protocol (MCP) and Agent-to-Agent (A2A) have been proposed to address interoperability challenges in LLM-based agent systems, enabling efficient tool access and effective multi-agent collaboration. **MCP**: Traditional Tool Use (TU) agents face significant challenges, including inconsistent APIs, high maintenance costs, and redundant development efforts, severely limiting interoperability across systems (Schick et al., 2023). To address these issues, Anthropic introduced the MCP, a unified communication layer allowing LLM-based agents to interact securely and consistently with external services and data sources via standardised interfaces. MCP mitigates data silo problems by providing dynamic service discovery and uniform access patterns. **A2A**: Google's A2A protocol facilitates decentralised multi-agent collaboration through structured, task-oriented dialogues. Agents from diverse vendors and model architectures can discover peers, delegate responsibilities, and collaboratively manage complex tasks as equal participants (Google, 2025). By abstracting agent discovery into Agent Cards and task coordination into Tasks and Artefacts, A2A supports flexible, incremental, multi-modal workflows, ideally suited to sophisticated collaborative scenarios.

MCP and A2A complement each other by clearly separating responsibilities: MCP serves as a standardised interface for accessing external tools, while A2A orchestrates collaborative agent interactions. Together, they establish a modular and scalable foundation for open, interoperable agent ecosystems, which can improve tool interoperability and inter-agent coordination in practical deployments.

## 3  Deep Research: Search Engine, Tool Use, Workflow, Tuning, Non-parametric Continual Learning

**Comparison with Conventional RAG-based Approaches.** DR agents expand the capabilities of traditional RAG methods by integrating dynamic retrieval, real-time TU, and adaptive reasoning into a unified system. RAG-based approaches typically rely on fixed pipelines, limiting their flexibility in handling complex, multi-step queries or rapidly changing contexts. In contrast, DR agents provide greater autonomy, context-awareness, and accuracy by dynamically engaging with external tools and managing multi-stage research tasks in real time.

In this section, we explore five core components essential for the development and optimization of DR agents: (3.1) **search engine integration**, which compares API-based interfaces with browser-based exploration to enhance dynamic knowledge acquisition; (3.2) **Tool Use capabilities**, which investigate the integration of code execution, mathematical computation, file manipulation, and multimodal processing modules within the agent's inference pipeline; (3.3) **workflow architecture**, analysing foundational designs, the balance between multi-agent and single-agent paradigms, memory mechanisms, and auxiliary components that facilitate the orchestration of complex research workflows; (3.4) **tuning methodologies**, which examine prompt-driven structured generation, LLM-driven prompting, fine-tuning strategies, and reinforcement learning approaches aimed at optimizing agent performance, and (3.5) **Non-parametric continual learning**, which enables LLM agents to self-evolve by dynamically adapting external tools, memory, and workflows without updating internal model weights, offering scalable optimization for complex tasks.

A cross-cutting observation across the following subsections is that retrieval, browsing, and general tool use can be understood as points along a unified tool invocation spectrum, differing primarily in execution complexity and environmental statefulness. API-based search (§3.1) represents lightweight, stateless tool calls; browser-based retrieval introduces stateful, multi-step interactions with dynamic content; and full computer use (§3.2), such as GUI navigation, constitutes the most complex form of environmental interaction. Recognising this shared abstraction suggests that future DR architectures may benefit from a unified tool dispatch layer that dynamically selects the appropriate invocation modality based on task requirements and cost constraints.

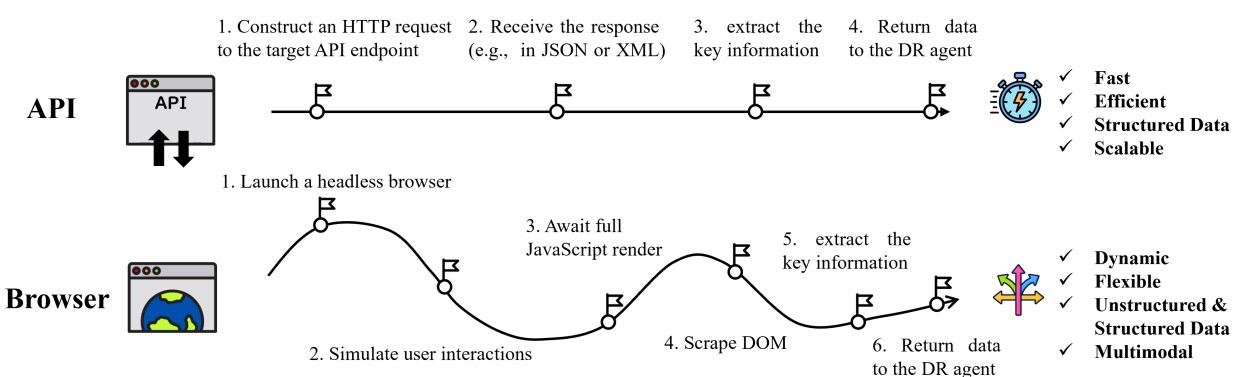

Figure 2: Comparison of DR Agents with Search Engine Details. Agents are listed chronologically and categorized by their primary retrieval modality (API-based vs. Browser-based).

### 3.1 Search Engine: API vs. Browser

To enhance reasoning depth and accuracy for handling evolving tasks, DR agents employ search engines (SE) to update their knowledge through interaction with the external environment. In Table 1, we present a comparative overview of SEs, base models, and evaluation benchmarks employed by existing DR agents. organised according to the retrieval modalities and implementation patterns discussed below. The SEs can be broadly categorised into two types:

1) **API-Based SEs**, which interact with structured data sources, such as search-engine APIs or scientific database APIs, enabling efficient retrieval of organised information.
2) **Browser-Based SEs**, which simulate human-like interactions with web pages, facilitating real-time extraction of dynamic or unstructured content, improving the comprehensiveness of the external knowledge.

Table 1: Comparison of DR Agents with Search Engine Details.

■ = **Primary focus**, ■ = **Unknown/Undisclosed**, □ = **Not present**

| DR Agent | Search Engine API | Browser | GAIA | HLE | Benchmark Other QA | Base Model | Release |
|---|---|---|---|---|---|---|---|
| *API-Based: Commercial/General Search APIs* | | | | | | | |
| Avatar Wu et al. (2024) | ■ | □ | □ | □ | Stark | Claude-3-Opus, GPT-4 | Feb-2024 |
| CoSearch-Agent Gong et al. (2024) | ■ | □ | □ | □ | □ | GPT-3.5-turbo | Feb-2024 |

*Continued on next page*

*Table 1 continued from previous page*

| DR Agent | Search Engine | | GAIA | HLE | Benchmark Other QA | Base Model | Release |
|---|---|---|---|---|---|---|---|
| | API | Browser | | | | | |
| MMAC-Copilot Song et al. (2024) | ■ | ☐ | ■ | ☐ | ☐ | GPT-3.5, GPT-4 | Mar-2024 |
| Storm Shao et al. (2024a) | ▨ | ☐ | ☐ | ☐ | FreshWiki | GPT-3.5-turbo | Jul-2024 |
| OpenResearcher Zheng et al. (2024) | ■ | ☐ | ☐ | ☐ | Privately Collected QA Data | DeepSeek-V2-Chat | Aug-2024 |
| Gemini DR Google Team (2025) | ■ | ■ | ☐ | ■ | GPQA | Gemini-2.0-Flash | Dec-2024 |
| Grok DeepSearch xAI Team (2025) | ■ | ■ | ☐ | ☐ | GPQA | Grok3 | Feb-2025 |
| Perplexity DR Perplexity Team (2025) | ■ | ▨ | ☐ | ■ | SimpleQA | Flexible | Feb-2025 |
| H2O.ai DR H2O.ai (2025) | ■ | ■ | ■ | ☐ | ☐ | h2ogpt-oasst1-512-12b | Mar-2025 |
| Suna AI AI (2025b) | ■ | ■ | ☐ | ☐ | ☐ | GPT-4o, Claude | Apr-2025 |
| Search-o1 Li et al. (2025b) | ■ | ☐ | ☐ | ☐ | GPQA, NQ, TriviaQA | QwQ-32B-preview | Jan-2025 |
| Cognitive Kernel-Pro Wan et al. (2025) | ■ | ☐ | ■ | ☐ | AgentWebQA, WebWalkerQA, Multi-hop URLQA, DocBench, TableBench | Claude-3.7-Sonnet, CK-Pro-8B | Aug-2025 |
| WideSearch Wong et al. (2025) | ■ | ☐ | ☐ | ☐ | WideSearch | DeepSeek-R1, Doubao-Seed-1.6, Claude Sonnet 4, Gemini-2.5-Pro | Aug-2025 |
| WebWatcher Geng et al. (2025) | ■ | ■ | ☐ | ■ | Browsercom-VL, LiveVQA, MMSearch | Qwen-2.5-VL-32B | Aug-2025 |
| ***API-Based: Specialized Academic/Scientific APIs*** | | | | | | | |
| The AI Scientist Lu et al. (2024) | ■ | ☐ | ☐ | ☐ | MLE-Bench | GPT-4o, o1-mini, o1-preview | Aug-2024 |
| Agent Laboratory Schmidgall et al. (2025) | ■ | ☐ | ☐ | ☐ | MLE-Bench | GPT-4o, o1-preview | Jan-2025 |
| Towards an AI co-scientist Gottweis et al. (2025) | ■ | ☐ | ☐ | ☐ | GPQA | Gemini 2.0 | Feb-2025 |
| Nouswise Nouswise (2025) | ☐ | ☐ | ☐ | ☐ | ☐ | — | Mar-2025 |
| AgentRxiv Schmidgall & Moor (2025) | ■ | ☐ | ☐ | ☐ | GPQA, MedQA | GPT-4o-mini | Mar-2025 |
| Agent-KB Tang et al. (2025b) | ■ | ☐ | ■ | ☐ | SWE-bench | GPT-4o, GPT-4.1, Claude-3.7-Sonnet, o3-mini, Qwen-3, DeepSeek-R1 | Jul-2025 |
| ***API-Based: RL-Optimized Search Policy*** | | | | | | | |
| Agentic Reasoning Wu et al. (2025c) | ■ | ☐ | ☐ | ☐ | GPQA | DeepSeek-R1, Qwen2.5 | Feb-2025 |
| R1-Searcher Song et al. (2025) | ▨ | ☐ | ☐ | ☐ | 2WikiMultiHopQA, HotpotQA | Llama3.1-8B-Inst, Qwen2.5-7B | Mar-2025 |
| ReSearch Chen et al. (2025b) | ▨ | ☐ | ☐ | ☐ | 2WikiMultiHopQA, HotpotQA | Qwen2.5-7B, Qwen2.5-7B-Inst | Mar-2025 |
| Search-R1 Jin et al. (2025) | ■ | ■ | ☐ | ☐ | 2WikiMultiHopQA, HotpotQA, NQ, TriviaQA | Llama3.2-3B, Qwen2.5-3B/7B | Mar-2025 |
| Agent-R1 Ouyang et al. (2025) | ■ | ☐ | ☐ | ☐ | HotpotQA | Qwen2.5-1.5B-Inst | Mar-2025 |
| OWL CAMEL-AI.org (2025) | ■ | ■ | ■ | ☐ | ☐ | Deepeek-R1, Gemini2.5-Pro, GPT-4o | Mar-2025 |
| SWIRL Goldie et al. (2025) | ■ | ☐ | ☐ | ☐ | HotQA, BeerQA | Gemma 2-27b | Apr-2025 |
| DeerFlow DanielWalnut (2025) | ■ | ☐ | ☐ | ☐ | ☐ | Doubao-1.5-Pro-32k, DeepSeek-R1, GPT-4o, Qwen | May-2025 |

*Table 1 continued from previous page*

| DR Agent | Search Engine API | Browser | GAIA | HLE | Benchmark Other QA | Base Model | Release |
|---|---|---|---|---|---|---|---|
| PANGU DEEP-DIVER Shi et al. (2025) | ■ | □ | □ | □ | C-SimpleQA, HotpotQA, ProxyQA | Pangu-7B-Reasoner | May-2025 |
| *Browser-Based: Full Environment Simulation* | | | | | | | |
| AutoAgent Tang et al. (2025a) | □ | ■ | ■ | □ | □ | Claude-Sonnet-3.5 | Feb-2025 |
| Manus Manus AI (2025) | ■ | ■ | □ | □ | □ | Claude3.5, GPT-4o | Mar-2025 |
| Openmanus Liang et al. (2025) | ■ | ■ | □ | □ | □ | Claude3.5, GPT-4o | Mar-2025 |
| Tool-Star Dong et al. (2025) | ■ | ■ | ■ | ■ | WebWalker, HotpotQA, 2WikiMultiHopQA | Qwen-2.5 | May-2025 |
| AgenticSeek Martin (2025) | □ | ■ | □ | □ | □ | GPT-4o, DeepSeek-R1, Claude | May-2025 |
| Alita Qiu et al. (2025) | ■ | ■ | ■ | □ | PathVQA | GPT-4o, Claude-Sonnet-4 | May-2025 |
| AWorld at InclusionAI (2025) | ■ | ■ | ■ | □ | HotpotQA | Gemini-2.5-Pro, GPT-4o | Jul-2025 |
| *Browser-Based: Navigation-Reasoning Loops* | | | | | | | |
| WebWalker Wu et al. (2025b) | □ | □ | □ | □ | WebWalkerQA | GPT-4o, Qwen-2.5 | Jan-2025 |
| AutoGLM Rumination Zhipu AI (2025) | □ | ■ | □ | □ | GPQA | GLM-Z1-Air | Mar-2025 |
| WebThinker Li et al. (2025c) | ■ | ■ | ■ | ■ | GPQA, WebWalkerQA | QwQ-32B | Apr-2025 |
| WebDancer Wu et al. (2025a) | ■ | ■ | ■ | □ | WebWalkerQA | Qwen-2.5, QwQ-32B, DeepSeek-R1, GPT-4o | May-2025 |
| O-agents Zhu et al. (2025a) | ■ | □ | ■ | □ | □ | GPT-4o, GPT-4.1, Claude-3.7-Sonnet, DeepSeek-R1, Gemini-2.5 | Jun-2025 |
| WebSailor Li et al. (2025a) | ■ | ■ | ■ | □ | SimpleQA | Qwen-2.5 | Jul-2025 |
| WebShaper Tao et al. (2025) | ■ | ■ | ■ | □ | WebWalkerQA | Qwen-2.5, QwQ-32B | Jul-2025 |
| MiroRL Team & Team (2025) | ■ | ■ | ■ | □ | □ | Qwen3-14B | Aug-2025 |
| *Browser-Based: Lightweight/Hybrid Fetching* | | | | | | | |
| DeepResearcher Zheng et al. (2025) | □ | ■ | ■ | □ | HotpotQA, NQ, TriviaQA | Qwen2.5-7B-Inst | Apr-2025 |
| SimpleDeepSearcher Sun* et al. (2025) | □ | ■ | ■ | □ | 2WikiMultiHopQA | Qwen-2.5-7B-In, Qwen-2.5-32B-In, DeepSeek-Distilled-Qwen-2.5-32B, QwQ-32B | Apr-2025 |
| Genspark Super Agent Team (2025b) | ■ | ■ | ■ | □ | □ | Mixture of Agents* | Apr-2025 |
| Kimi-Researcher Moonshot AI (2025) | ■ | ■ | □ | ■ | SimpleQA | Kimi k1.5/k2 | Jun-2025 |
| *Proprietary Systems (Implementation Undisclosed)* | | | | | | | |
| OpenAI DR OpenAI (2025b) | □ | ■ | ■ | ■ | ■ | GPT-o3 | Feb-2025 |
| Copilot Researcher Microsoft (2025) | □ | ■ | □ | □ | □ | o3-mini | Mar-2025 |
| Deep Researcher with Test-Time Diffusion Han et al. (2025) | ■ | □ | ■ | ■ | □ | Gemini-2.5-Pro | Jul-2025 |
| ChatGPT-Agent OpenAI (2025a) | □ | □ | □ | □ | □ | — | Jul-2025 |

**API-Based Retrieval.** API-based retrieval prioritises scalability and structured access through three primary implementation patterns. Most agents leverage commercial interfaces such as Google, Bing, or Serper for broad coverage: Gemini DR (Google Team, 2025) and Grok DeepSearch (xAI Team, 2025) coordinate multiple feeds including news and Wikipedia to decompose queries, while Perplexity DR (Perplexity Team, 2025) and Cognitive Kernel-Pro (Wan et al., 2025) aggregate results for comprehensive reporting. Similarly, Avatar, CoSearch-Agent, Storm, MMAC-Copilot, OpenResearcher, H2O.ai DR, Suna AI, and WideSearch (Wu et al., 2024; Gong et al., 2024; Shao et al., 2024a; Song et al., 2024; Zheng et al., 2024; H2O.ai, 2025; AI, 2025b; Wong et al., 2025) utilise general search APIs to ground open-domain QA and content generation, with systems like Search-o1 and WebWatcher (Li et al., 2025b; Geng et al., 2025) adding reader APIs or multimodal processing. Moving beyond general search, specialized agents target structured academic repositories: Agent Laboratory, AgentRxiv, Agent-KB, and Towards an AI co-scientist (Schmidgall et al., 2025; Schmidgall & Moor, 2025; Tang et al., 2025b; Gottweis et al., 2025) automate literature reviews via the arXiv API, while The AI Scientist (Lu et al., 2024), DeepRetrieval (Jiang, 2025), and Nouswise (Nouswise, 2025) validate novelty or query biomedical databases. Finally, recent frameworks treat search as a learnable policy: Agentic Reasoning, ReSearch, R1-Search, and Search-R1 (Wu et al., 2025c; Chen et al., 2025b; Song et al., 2025; Jin et al., 2025) employ reinforcement learning to optimise query timing, a paradigm extended by SWIRL, PANGU DeepDiver, Agent-R1, OWL, and DeerFlow (Goldie et al., 2025; Shi et al., 2025; Ouyang et al., 2025; CAMEL-AI.org, 2025; DanielWalnut, 2025) to adapt search intensity to task difficulty.

**Browser-Based Retrieval.** Complementing APIs, browser-based methods enable interaction with dynamic, unstructured, or deeply nested web content through three architectural approaches. The first involves full environment simulation, where agents operate sandboxed instances (e.g., Chromium) to emulate human interaction. Manus (Manus AI, 2025) and OpenManus (Liang et al., 2025) execute programmatic sessions to handle forms and lazy loading, while AutoAgent and AgenticSeek (Tang et al., 2025a; Martin, 2025) function within environment wrappers for stealthy navigation. Similarly, Tool-Star, Alita, and AWorld (Dong et al., 2025; Qiu et al., 2025; at InclusionAI, 2025) isolate browsing as a distinct module for complex multi-step tasks. A second approach tightly couples navigation with reasoning via sequential action-observation loops: WebThinker, WebWalker, WebDancer, WebSailor, and WebShaper (Li et al., 2025c; Wu et al., 2025b;a; Li et al., 2025a; Tao et al., 2025) treat page interactions as discrete observations to iteratively drill down into content, a strategy further refined by AutoGLM's (Zhipu AI, 2025) rumination cycles and the RL-integrated policies of MiroRL and O-agents (Team & Team, 2025; Zhu et al., 2025a). The third category employs lightweight fetching or hybrid workflows to balance depth with cost: SimpleDeepSearcher and Kimi-Researcher (Sun* et al., 2025; Moonshot AI, 2025) utilise text-based browsers or HTTP fetching, DeepResearcher (Zheng et al., 2025) processes webpages in segments, and Genspark Super Agent (Team, 2025b) distributes reading tasks across specialized sub-agents. Regarding proprietary systems such as OpenAI DR and Copilot Researcher (OpenAI, 2025b; Microsoft, 2025), while implementation details remain undisclosed, their observed capability, their observed capability to handle interactive widgets and multi-step navigation suggests that they may employ architectures functionally similar to these browser-based approaches, though this remains unverified due to the absence of public documentation.

**Strategic Trade-offs and Guidelines.** The choice between API and browser modalities represents a fundamental architectural trade-off. **API-based methods** are preferable for high-throughput, budget-constrained scenarios where latency is critical, yet they suffer from *information flattening*, losing the structural context of the source. Conversely, **Browser-based methods** are essential for deep research requiring interaction with dynamic DOM elements or gated content, but they introduce significant latency, higher computational costs, and stability challenges due to anti-bot measures. A growing challenge across

both modalities is *over-retrieval* and *diminishing returns*, where agents accumulate redundant information that dilutes reasoning context. Consequently, modern designs increasingly adopt hybrid router systems that dynamically select the retrieval modality based on query complexity.

## 3.2 Tool Use: Empowering Agents with Extended Functionalities

To expand DR agents' capacity to interact with external environments in complex research tasks, specifically by actively invoking and handling diverse tools and data sources, various DR agents have introduced three core tool modules: code interpreters, data analytics, multimodal processing, along with the Model Context Protocol. Tabel 2 provides a comparative overview of tool use capabilities across DR agents.

**Computational Reasoning and Environment Interaction.** To circumvent the inherent limitations of LLMs regarding precise calculation and complex logic, code interpreters serve as the central reasoning engine. Storm, The AI Scientist, Agent Laboratory, Agentic Reasoning, and Agent-R1 (Shao et al., 2024a; Lu et al., 2024; Schmidgall et al., 2025; Wu et al., 2025c; Ouyang et al., 2025) leverage Python execution to structure the research process, manage citations, and verify logically complex claims. In the open-source domain, OpenManus, Tool-Star, AgenticSeek, AutoAgent, DeerFlow, Suna AI, and MiroRL (Liang et al., 2025; Dong et al., 2025; Martin, 2025; Tang et al., 2025a; DanielWalnut, 2025; AI, 2025b; Team & Team, 2025) implement code-based modules to parse retrieved data and execute multi-step workflows. Alita (Qiu et al., 2025) extends this by exploring self-evolving agents that generate and wrap MCP tools with minimal predefined schemas. This execution paradigm is rapidly evolving into comprehensive "Computer Use", where agents autonomously control user interfaces. A prominent example is AutoGLM Rumination (Zhipu AI, 2025), which operationalises this approach by interacting with authenticated environments such as the China National Knowledge Infrastructure (CNKI) and WeChat Official Accounts, thereby enabling tasks that require session management and effectively bridging the gap between abstract reasoning and authentic environment interaction.

**Data Synthesis and Analytics.** For tasks necessitating the aggregation of large-scale information or the generation of visual artifacts, agents integrate dedicated analytics pipelines. Proprietary systems such as OpenAI DR, Grok DeepSearch, Perplexity DR, Genspark DR, and Manus (OpenAI, 2025b; xAI Team, 2025; Perplexity Team, 2025; Team, 2025b; Manus AI, 2025) appear to integrate code interpreters with specialised libraries for statistical analyses and chart generation. As these systems do not disclose internal architectures, this inference is drawn from observable outputs rather than verified implementation details. A similar capability is also observed in Copilot Researcher and H2O.ai DR (Microsoft, 2025; H2O.ai, 2025). Among open frameworks, WebThinker, Kimi-Researcher, OWL, and AWorld (Li et al., 2025c; Moonshot AI, 2025; CAMEL-AI.org, 2025; at InclusionAI, 2025) provide dedicated analytics tools to synthesise diverse web sources. Notably, CoSearchAgent and Towards an AI co-scientist (Gong et al., 2024; Gottweis et al., 2025) focus specifically on data-driven insights, utilising analytics modules to process team inputs or scientific data without necessarily relying on arbitrary code generation for all tasks.

**Multimodal Perception and Generation.** To handle the visual nature of web content and scientific documents, complex agents incorporate multimodal processing. Cognitive Kernel-Pro and WebWatcher (Wan et al., 2025; Geng et al., 2025) explicitly integrate vision-language models to interpret charts, diagrams, and webpage layouts. O-agents and Agent-KB (Zhu et al., 2025a; Tang et al., 2025b) leverage these capabilities

---

*Mixture of Agents refers to an ensemble of nine base models comprising GPT-4.1, GPT-o3, GPT-o4-mini-high, Claude-Sonnet-3.7-Thinking, Claude-Sonnet-3.7, Gemini-2.0-Flash, Gemini-2.5-Pro, DeepSeek-V3, DeepSeek-R1

Table 2: Comparison of DR Agents with Tool Use Capabilities. Agents are grouped by their primary functional focus as discussed in Section 3.2: Computational Reasoning (code execution and environment interaction), Data Synthesis (analytics pipelines), and Multimodal Processing (vision-language integration). Within each group, agents are sorted chronologically.

■ = Involved, ■ = Undisclosed/Unknown, □ = Not present

| DR Agent | Code Interpreter | Data Analytics | Multimodal | Release |
|---|:---:|:---:|:---:|:---:|
| **_Computational Reasoning and Environment Interaction_** | | | | |
| Storm Shao et al. (2024a) | ■ | □ | □ | Jul-2024 |
| The AI Scientist Lu et al. (2024) | ■ | □ | □ | Aug-2024 |
| Agent Laboratory Schmidgall et al. (2025) | ■ | □ | □ | Jan-2025 |
| Agentic Reasoning Wu et al. (2025c) | ■ | □ | □ | Feb-2025 |
| Agent-R1 Ouyang et al. (2025) | ■ | □ | □ | Mar-2025 |
| OpenManus Liang et al. (2025) | ■ | ■ | □ | Mar-2025 |
| MiroRL (Team & Team, 2025) | ■ | ■ | □ | Aug-2025 |
| **_Data Synthesis and Analytics_** | | | | |
| CoSearchAgent Gong et al. (2024) | □ | ■ | □ | Feb-2024 |
| Towards an AI co-scientist (Gottweis et al., 2025) | □ | ■ | ■ | Feb-2025 |
| OWL CAMEL-AI.org (2025) | ■ | ■ | ■ | Mar-2025 |
| WebThinker Li et al. (2025c) | ■ | ■ | □ | Apr-2025 |
| Suna Ai AI (2025b) | ■ | ■ | □ | Apr-2025 |
| Tool-Star (Dong et al., 2025) | ■ | ■ | □ | May-2025 |
| AgenticSeek Martin (2025) | ■ | ■ | □ | May-2025 |
| DeerFlow DanielWalnut (2025) | ■ | ■ | □ | May-2025 |
| Kimi-Researcher (Moonshot AI, 2025) | ■ | ■ | □ | Jun-2025 |
| AWorld (at InclusionAI, 2025) | ■ | ■ | ■ | Jul-2025 |
| **_Multimodal Perception and Generation_** | | | | |
| AutoAgent Tang et al. (2025a) | ■ | □ | ■ | Feb-2025 |
| AutoGLM Rumination Zhipu AI (2025) | ■ | □ | ■ | Mar-2025 |
| O-agents (Zhu et al., 2025a) | ■ | ■ | ■ | Jun-2025 |
| Agent-KB (Tang et al., 2025b) | ■ | ■ | ■ | Jul-2025 |
| Cognitive Kernel-Pro (Wan et al., 2025) | ■ | ■ | ■ | Aug-2025 |
| WebWatcher (Geng et al., 2025) | ■ | ■ | ■ | Aug-2025 |
| **_Proprietary Systems (Implementation Undisclosed)_** | | | | |
| Genspark DR Team (2025b) | ■ | ■ | ■ | Feb-2025 |
| Grok DeepSearch xAI Team (2025) | ■ | ■ | ■ | Feb-2025 |
| OpenAI DR OpenAI (2025b) | ■ | ■ | ■ | Feb-2025 |
| Perplexity DR Perplexity Team (2025) | ■ | ■ | ■ | Feb-2025 |
| Copilot Researcher Microsoft (2025) | ■ | ■ | ■ | Mar-2025 |
| Manus Manus AI (2025) | ■ | ■ | ■ | Mar-2025 |
| H2O.ai DR H2O.ai (2025) | ■ | ■ | ■ | Mar-2025 |
| Genspark Super Agent Team (2025b) | ■ | ■ | ■ | Apr-2025 |
| Alita Qiu et al. (2025) | ■ | ■ | ■ | May-2025 |

to construct knowledge bases from visual sources. Whilst implementation specifics for Alita and Copilot Researcher remain undisclosed, major platforms including Genspark Super Agent, Grok, OpenAI DR, and Gemini-based systems exhibit multimodal capabilities in their publicly available outputs, suggesting capability in parsing PDFs and analysing visual data alongside text, though the underlying architectures remain proprietary.

## 3.3 Architecture and Workflow

As shown in Figure 4, this section systematically analyses the construction of DR systems, focusing on workflows categorised into **static** and **dynamic** types. We first introduce the static workflows and then discuss planning strategies, which enhance task allocation and execution through **three distinctive user interaction types** to clarify intent: planning-only (direct planning without clarifying user intent), intent-to-planning (clarifying intent before planning to align the task with user goals), and unified intent-planning, which simultaneously generates an initial research plan from the user's prompt and interactively engages the user to confirm, modify, or reject the proposed plan before execution begins. The distinction between **single-agent** and **multi-agent** systems is examined in the context of dynamic workflows, emphasising specialisation in task management. Additionally, we examine memory mechanisms for managing and integrating retrieved information, which enhance the performance and adaptability of DR systems.

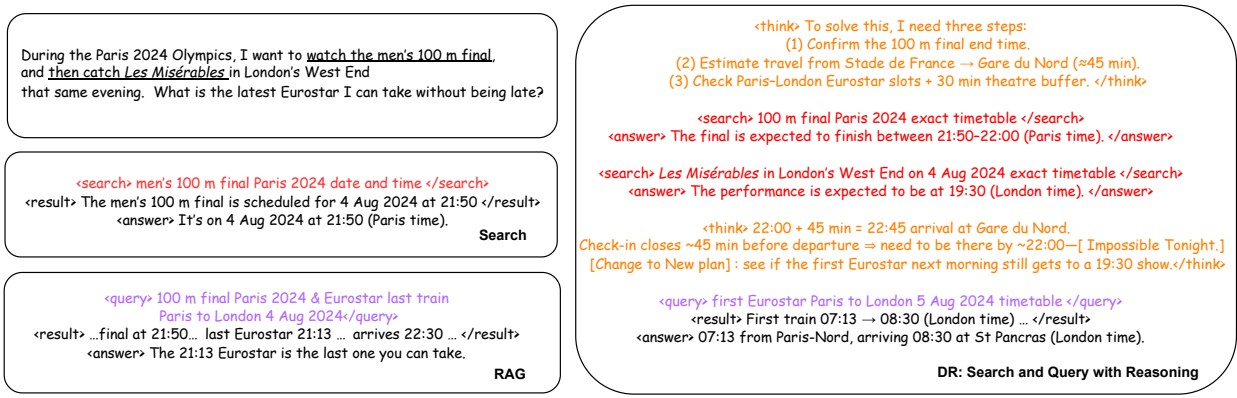

Figure 3: **Comparison of Information Retrieval Methods.** The upper left corner (Search) represents the searching methods, which can use the browser or API; the lower left corner (RAG, Query) represents Retrieval-Augmented Generation, combining retrieval and generative models to output natural language answers; the right side (Deep Research) represents the deep research process, generating complex decisions or analyses through retrieval and explicit reasoning.

### 3.3.1 Static vs. Dynamic Workflows

**Static Workflows.** Static workflows rely on manually predefined task pipelines, decomposing research processes into sequential subtasks executed by dedicated agents. These workflows follow explicitly structured procedures, making them particularly suitable for well-defined, structured research scenarios. For instance, AI Scientist (Lu et al., 2024) automates scientific discovery through distinct sequential phases, including ideation, experimentation, and reporting. Similarly, Agent Laboratory (Schmidgall et al., 2025) segments

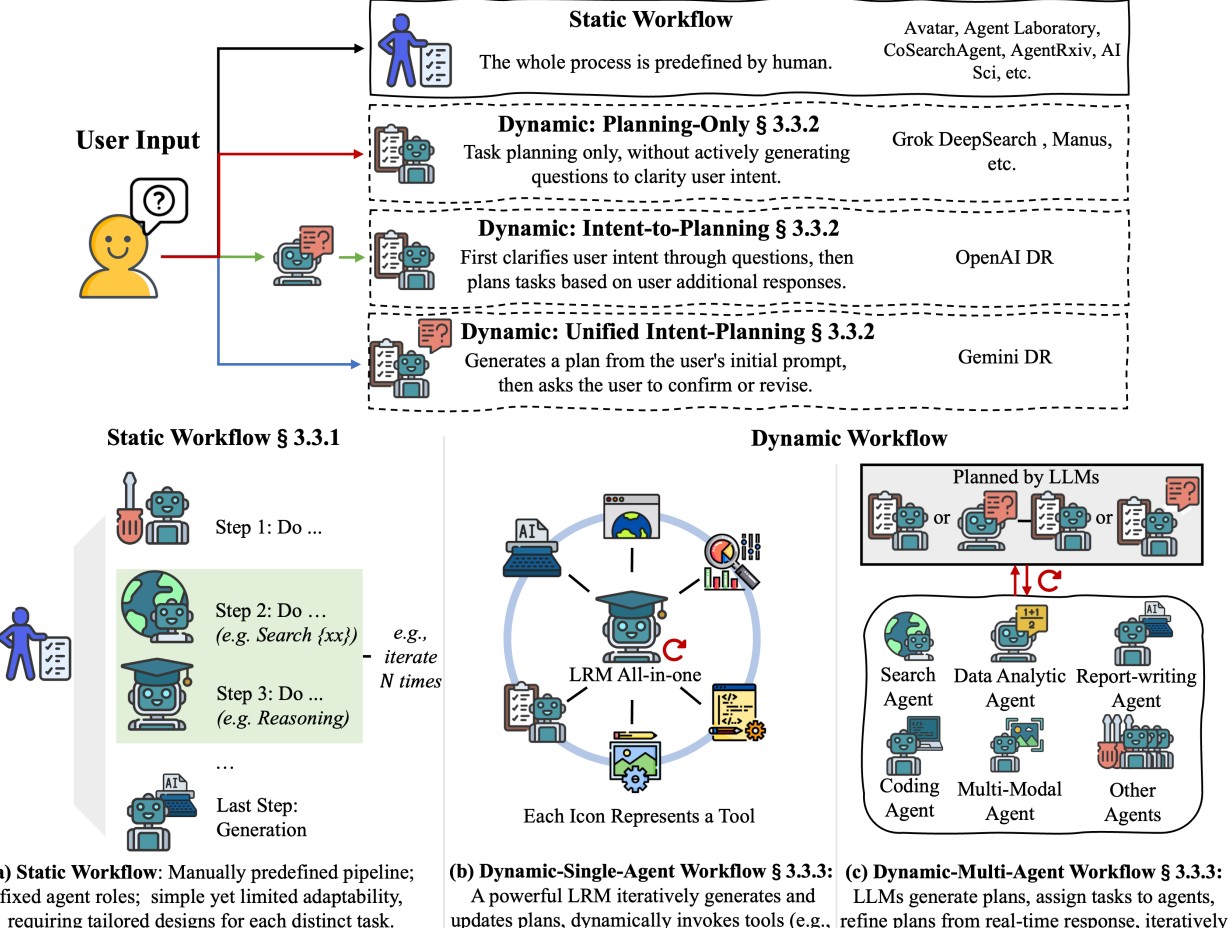

Figure 4: Comparison of DR Workflows: (1) **Static vs. Dynamic Workflows**: Static workflows rely on predefined task sequences, while dynamic workflows allow LLM-based task planning. (2) **Planning Strategies**: Three types include: planning-only (direct planning without clarifying user intent), intent-to-planning (clarifying intent before planning), and unified intent-planning (generating a plan and requesting user confirmation). (3) **Single-Agent vs. Multi-Agent**: Dynamic workflows can be categorised into dynamic-multi-agent systems (tasks distributed across specialised agents) or dynamic-single-agent systems (an LRM autonomously updates and executes tasks).

research activities into formalised stages, such as literature review, experimentation, and synthesis of findings. Extending this static paradigm further, AgentRxiv (Schmidgall & Moor, 2025) incorporates inter-agent collaboration mechanisms, enabling incremental knowledge reuse through sharing intermediate research outcomes among specialised agents. Figure 3 illustrates the differences among search, RAG, and DR in handling a complex multi-step query.

Despite their ease of implementation and structured clarity, static workflows suffer from limited generalisation capabilities, as each distinct task necessitates a specifically tailored pipeline.

**Dynamic Workflows.** To overcome the limitations in flexibility and generalizability inherent in static workflows, dynamic workflows support adaptive task planning, allowing agents to dynamically reconfigure task structures based on iterative feedback and evolving contexts. Dynamic architectures leverage advanced mechanisms including automated planning, iterative refinement, and interactive task allocation, enabling tasks to evolve in real-time as new knowledge or external inputs become available. Consequently, dynamic workflows exhibit greater generality and adaptability, making them highly suitable for complex, knowledge-intensive tasks commonly encountered in AI-driven research scenarios.

### 3.3.2 Dynamic Workflows: Planning Strategies

To enhance DR agents' adaptability in response to evolving user requirements and contexts, existing studies propose three distinctive LLM-based planning strategies, each differing in whether and how they interact with the user to clarify intent:

1) The **Planning-Only** approach directly generates task plans based solely on initial user prompts without actively engaging in further clarification, adopted by the majority of existing DR agents, including Grok (xAI Team, 2025), H2O (H2O.ai, 2025) and Manus (Manus AI, 2025).
2) The **Intent-to-Planning** strategy actively clarifies user intent prior to planning through targeted questions, subsequently generating tailored task sequences based on clarified user inputs; this method is utilised by OpenAI DR (OpenAI, 2025b).
3) The **Unified Intent-Planning** approach synthesises these methods by generating a preliminary plan from the initial prompt, together with interactively engaging the user to confirm or revise the proposed plan. Gemini DR (Google Team, 2025) is representative of this strategy, effectively adopts the strength of user-guided refinement.

### 3.3.3 Dynamic Workflows: Single-Agent vs. Multi-Agent

Dynamic workflows of DR agents can be differentiated based on agent architectures into single-agent and multi-agent frameworks, each exhibiting distinct characteristics concerning task specialisation, coordination complexity, and scalability of execution.

**Dynamic Single-Agent Systems.** Dynamic single-agent systems **integrate planning, tool invocation, and execution within a unified LRM**, streamlining task management into a cohesive cognitive loop. Single-agent architectures autonomously refine task plans and invoke appropriate tools based on evolving contexts, typically without explicit inter-agent coordination. Compared to multi-agent architectures, single-agent systems enable direct end-to-end reinforcement learning (RL) optimisation across the entire workflow, facilitating smoother and more coherent integration of reasoning, planning, and tool invocation. Systems such as Search-o1 (Li et al., 2025b), R1-Searcher (Song et al., 2025), DeepResearcher (Zheng et al., 2025), WebDancer (Wu et al., 2025a), WebSailor (Li et al., 2025a), PANGU Deepdiver (Shi et al., 2025), Agent-

R1 (Ouyang et al., 2025), ReSearch (Chen et al., 2025b), Search-R1 (Jin et al., 2025), WebWatcher (Wu et al., 2025b), MiroRL (Team & Team, 2025) and Kimi-Researcher (Moonshot AI, 2025) exemplify this paradigm through iterative cycles of explicit reasoning, action, and reflection, aligning with the ReAct framework (Yao et al., 2023). However, this streamlined approach places significant demands on the foundation model's reasoning capabilities, contextual understanding, and autonomous selection and invocation of tools. Additionally, the tightly integrated nature of single-agent systems may limit modular flexibility, complicating independent scaling or optimisation of individual functional components.

**Dynamic Multi-Agent Systems.** Dynamic multi-agent systems leverage multiple specialised agents to collaboratively execute subtasks generated and dynamically allocated through adaptive planning strategies. These systems typically employ hierarchical or centralised planning mechanisms, wherein a coordinator agent continuously assigns and redistributes tasks based on real-time feedback and replanning. Representative frameworks include OpenManus (Liang et al., 2025) and Manus (Manus AI, 2025), both adopting hierarchical planner-toolcaller architectures. Similarly, OWL (CAMEL-AI.org, 2025) includes a workforce-oriented model, utilising a central manager agent to orchestrate task distribution among specialised execution agents. Furthermore, Alita (Qiu et al., 2025) incorporates a self-evolution mechanism into DR agents, allowing the agent to online instantiate and configure new MCP servers tailored to specific tasks and environmental conditions. AWorld (at InclusionAI, 2025) is an open-source framework for building, orchestrating, and training tools using agents and larger multi-agent systems, offering memory and context services and MCP tool integration for scalable evaluation and self-improvement. Webwalker (Wu et al., 2025b) mimics human-like web navigation through an explore-critic paradigm. WebThinker (Li et al., 2025c) uses both executed and auxiliary agents to autonomously search, deeply explore web pages, and draft research reports. Such multi-agent configurations effectively handle complex, parallelizable research tasks, thereby enhancing flexibility and scalability in open-ended research scenarios. Nevertheless, a major current challenge of multi-agent systems lies in the inherent complexity of coordinating multiple independent agents, making it difficult to conduct effective end-to-end reinforcement learning optimisation. A notable architectural distinction between the two paradigms lies in tool and protocol integration. In single-agent systems, all tool invocations, including MCP connections are managed through a unified interface, simplifying protocol management but creating potential context-window bottlenecks. In multi-agent systems, tool access can be distributed: each specialised agent may maintain its own MCP connections (e.g., a search agent with search API servers, a coding agent with interpreter servers), which improves modularity but requires explicit coordination protocols to manage shared state and prevent conflicting tool invocations. This distinction has implications for both scalability and the credit assignment challenge in end-to-end optimisation.

### 3.3.4 Memory Mechanism for Long-Context Optimisation

Memory mechanisms empower DR agents to persistently capture, organise, and recall relevant information across multiple retrieval rounds, thereby reducing redundant queries and improving both the efficiency and coherence of DR tasks. During the DR process, agents typically perform extensive multi-round retrieval, generating hundreds of thousands of tokens (or even millions). Although recent advances in LLMs have significantly expanded context window sizes, current limits still constrain tasks involving extremely long contexts. To address these challenges, DR systems have implemented various optimisations for processing extended contexts. Broadly, these optimisations can be categorised into three main strategies: **(i) Expanding the Context Window Length; (ii) Compressing Intermediate Steps; (iii) Utilising External Structured Storage for Temporary Results**.

**Extending the Context Window Length.** It is the simple but intuitively effective approach, exemplified by Google's Gemini model (Google Team, 2025), which supports a context window of up to one million tokens, supplemented by a RAG setup. Despite its straightforwardness, this method often incurs high computational costs and may lead to inefficiencies in resource utilisation during practical deployments.

**Compressing Intermediate Steps.** An alternative strategy involves compressing or summarising intermediate reasoning steps, significantly reducing the number of tokens processed by the model and thereby improving both efficiency and output quality. Representative frameworks such as The AI Scientist (Lu et al., 2024) and CycleResearcher (Weng et al., 2024) pass summarised intermediate results between workflow phases. Further, Search-o1 (Li et al., 2025b) introduced the concept of "Reason-in-Documents", utilising LRMs to compress documents, substantially reducing token volume and enhancing model decision-making efficiency. Meanwhile, WebThinker (Li et al., 2025c) uses an auxiliary model to compress the external information. However, a potential drawback of this approach is the loss of detailed information, potentially impacting the precision of subsequent reasoning.

**Utilising External Structured Storage.** This is for preserving and retrieving historical information, enabling DR agents to persistently and efficiently store vast amounts of past context beyond the constraints of the context window, improving memory capacity, retrieval speed, and semantic relevance. Popular open-source frameworks such as Manus (Manus AI, 2025), OWL (CAMEL-AI.org, 2025), Open Manus (Liang et al., 2025), and Avatar (Wu et al., 2024) utilise external file systems to store intermediate outcomes and historical data for subsequent retrieval. Frameworks like AutoAgent (Tang et al., 2025a) have developed self-managing modules that leverage vector databases to support scalable memory storage and fast similarity-based lookup. Beyond plain text or vector stores, some works propose more semantically structured memory frameworks: for instance, Agentic Reasoning (Wu et al., 2025c) employ knowledge graphs to capture intermediate reasoning processes and thereby enhance the precision of information reuse, while Agentrxiv (Schmidgall & Moor, 2025) simulates an academic repository akin to arXiv for storing and retrieving relevant outcomes from other agents. Furthermore, Agent-KB (Tang et al., 2025b) and Alita (Qiu et al., 2025) construct shared knowledge bases and optimised toolsets for agentic problem-solving. Although these structured approaches offer superior semantic retrieval efficiency and accuracy, they typically entail higher development and maintenance costs due to the need for meticulous data structure design and management.

### 3.4 Tuning: Beyond Prompting toward Capability Enhancement

Table 3: Comparison of DR Agents with Tuning Methods

**Symbols:** ■ = Yes, ▨ = Undisclosed, □ = Not present.
**Paradigms: Hybrid** (SFT+RL); **Direct** (RL w/o SFT); **SFT-Only** (No RL); **Prompting** (Frozen Weights); **Proprietary** (Closed).

| Agent | SFT | RL | Paradigm | Base Model | Data | Reward |
|---|---|---|---|---|---|---|
| *Paradigm: Proprietary* | | | | | | |
| Gemini DR Google Team (2025) | ▨ | ▨ | Proprietary | Gemini-2.0 | □ | ▨ |
| Grok DeepSearch | □ | ▨ | Proprietary | Grok3 | □ | ▨ |
| OpenAI DR OpenAI (2025b) | □ | ▨ | Proprietary | GPT-o3 | □ | ▨ |
| AutoGLM | ▨ | ▨ | Proprietary | GLM-Z1 | □ | Process-based |
| Copilot Researcher | ▨ | ▨ | Proprietary | o3-mini | □ | □ |
| Nouswise (Nouswise, 2025) | ▨ | ▨ | Proprietary | Nouswise | □ | ▨ |

*Table 3 continued*

| Agent | SFT | RL | Paradigm | Base Model | Data | Reward |
|---|---|---|---|---|---|---|
| Genspark Agent | ☐ | ▨ | Proprietary | MoA | ☐ | ▨ |
| ***Paradigm: Prompting*** | | | | | | |
| WebWalker (Wu et al., 2025b) | ☐ | ☐ | Prompting | GPT-4o | WebWalker | ☐ |
| AI co-scientist | ☐ | ☐ | Prompting | Gemini 2.0 | ☐ | ☐ |
| O-agents | ☐ | ☐ | Prompting | GPT-4o | ☐ | ☐ |
| Agent-KB (Tang et al., 2025b) | ☐ | ☐ | Prompting | GPT-4o | ☐ | ☐ |
| ***Paradigm: SFT-Only*** | | | | | | |
| Agentic Reasoning Wu et al. (2025c) | ■ | ☐ | SFT-Only | DeepSeek-R1 | GPQA | Rule-Outcome |
| H2O.ai DR H2O.ai (2025) | ■ | ▨ | SFT-Only | h2ogpt | ☐ | ▨ |
| Cognitive Kernel | ■ | ☐ | SFT-Only | Claude-3.7 | OpenWeb | ☐ |
| ***Paradigm: Direct (RL w/o SFT)*** | | | | | | |
| AutoAgent | ☐ | ■ | Direct | Claude-3.5 | ☐ | ☐ |
| Agent-R1 Ouyang et al. (2025) | ☐ | PPO, GRPO | Direct | Qwen2.5 | HotpotQA | Rule-Outcome |
| ReSearch Chen et al. (2025b) | ☐ | GRPO | Direct | Qwen2.5 | 2Wiki | Rule-Outcome |
| DeepResearcher | ☐ | GRPO | Direct | Qwen2.5 | NQ | Rule-Outcome |
| SWIRL Goldie et al. (2025) | ☐ | Offline-RL | Direct | Gemma-2 | HotPotQA | ☐ |
| Kimi-Researcher | ☐ | REINFORCE | Direct | Kimi k1.5 | ☐ | Rule-Outcome |
| WebWatcher | ☐ | GRPO | Direct | Qwen-VL | BrowseComp | Rule-Outcome |
| ***Paradigm: Hybrid (SFT + RL)*** | | | | | | |
| R1-Searcher Song et al. (2025) | ■ | GRPO, RF++ | Hybrid | Qwen2.5 | 2Wiki | Rule-Outcome |
| Search-R1 Jin et al. (2025) | ■ | PPO, GRPO | Hybrid | Qwen2.5 | NQ | Rule-Outcome |
| WebThinker | ■ | DPO | Hybrid | QwQ-32B | ExpertData | Rule-Outcome |
| SimpleDeepSearcher | ■ | PPO | Hybrid | Qwen-2.5 | NQ | Process-based |
| PANGU DEEPDIVER | ■ | GRPO | Hybrid | Pangu-7B | WebPuzzle | Rule-Outcome |
| Tool-Star (Dong et al., 2025) | ■ | GRPO | Hybrid | Qwen-2.5 | NuminaMath | Rule-Outcome |
| WebDancer | ■ | DAPO | Hybrid | Qwen-2.5 | CRAWLQA | Rule-Outcome |
| WebSailor | ■ | DuPO | Hybrid | Qwen-2.5 | SailorFog | Rule-Outcome |
| WebShaper | ■ | GRPO | Hybrid | Qwen-2.5 | WebShaper | Rule-Outcome |
| MiroRL (Team & Team, 2025) | ■ | GRPO | Hybrid | Qwen3 | MiroRLQA | Rule-Outcome |

**Parametric Approaches.** While prompt-based methods enable rapid prototyping by leveraging frozen pre-trained weights, they impose inherent ceilings on reasoning depth and stability when handling complex research workflows. To circumvent these limitations, the field has shifted towards parametric optimisation to internalise specialised behaviours such as dynamic tool usage and long-horizon planning. This section examines the two primary paradigms driving this transition: Supervised Fine-Tuning (SFT), which bootstraps fundamental research capabilities from expert demonstrations, and Reinforcement Learning (RL), which further refines search strategies and robustness through outcome-driven or process-driven optimisation.

### 3.4.1 Supervised Fine-Tuning (SFT)

Supervised Fine-Tuning (SFT) serves as the foundational step for equipping agents with domain-specific tool usage, search syntax, and structured reasoning capabilities. While general approaches like Open-RAG and AUTO-RAG (Islam et al., 2024; Yu et al., 2024b) demonstrated the utility of tuning on retrieval tokens, modern DR agents adapt this specifically for complex, long-horizon workflows.

The most prevalent application of SFT involves initialising the agent's policy using demonstrations collected from expert trajectories or curated benchmark datasets. Specifically, agents such as Agentic Reasoning, H2O.ai DR, SimpleDeepSearcher, R1-Searcher, and Search-R1 (Wu et al., 2025c; H2O.ai, 2025; Sun* et al., 2025; Song et al., 2025; Jin et al., 2025) utilise datasets derived from GPQA, HotpotQA, or 2WikiMultiHopQA, where expert-annotated sequences of query decomposition, search action, and answer synthesis serve as supervision signals to teach the model how to decompose queries and format tool calls. This phase is crucial for stabilising output formats and grounding the model's initial policy. Similarly, WebThinker, Tool-Star, and WebSailor (Li et al., 2025c; Dong et al., 2025; Li et al., 2025a) leverage expert datasets or specific QA pairs (e.g., SailorFog-QA) to align the base model with optimal research behaviours.

Beyond standard benchmarks, several frameworks construct specialised datasets to target distinct capabilities. WebShaper (Tao et al., 2025) trains on its proprietary WebShaper dataset to enhance layout understanding. Cognitive Kernel-Pro (Wan et al., 2025) employs a diverse mixture of OpenWebVoyager, DocBench, and TableBench to fine-tune its multimodal handling. Furthermore, WebDancer, PANGU DEEPDIVER, and MiroRL (Wu et al., 2025a; Shi et al., 2025; Team & Team, 2025) incorporate SFT as a precursor to reinforcement learning, ensuring the model's action space is well-defined before exploration.

In contrast, a subset of agents relies exclusively on the intrinsic generalisation capabilities of strong base models without task-specific fine-tuning. WebWalker, Towards an AI co-scientist, O-agents, and Agent-KB (Wu et al., 2025b; Gottweis et al., 2025; Zhu et al., 2025a; Tang et al., 2025b) operate zero-shot or few-shot, leveraging architectures such as GPT-4o, Gemini 2.0, or DeepSeek-R1 to perform reasoning directly from prompts.

### 3.4.2 Reinforcement Learning (RL)

Reinforcement Learning enables agents to adapt to open-ended research environments by optimising reasoning paths based on outcome or process rewards. To facilitate this, a diverse array of algorithms has emerged. Classical methods such as *REINFORCE* (utilised by Kimi-Researcher (Moonshot AI, 2025)) and *Proximal Policy Optimisation (PPO)* (Schulman et al., 2017) provide foundational stability via policy gradients and value networks. To reduce memory overhead and leverage group dynamics, *Group Relative Policy Optimisation (GRPO)* (Shao et al., 2024b) and *Reinforce++* (Hu, 2025) eliminate separate value networks by normalising advantages within response groups. For offline or preference-based alignment, *Direct Preference Optimisation (DPO)* (Rafailov et al., 2023) and *Offline-RL* allow learning from static datasets, whilst specialised variants like *Dual Policy Optimisation (DuPO)* She et al. (2025) and *Direct Alignment from Predictive Output (DAPO)* (Yu et al., 2025) target specific trade-offs in reasoning stability and trajectory consistency. Leveraging these algorithms, our analysis of Table 3 reveals five distinct implementation patterns.

The first is the **Hybrid Paradigm** (*Hybrid*), combining SFT bootstrapping with policy refinement. Agents like SimpleDeepSearcher, Search-R1, Tool-Star, WebShaper, and WebThinker (Sun* et al., 2025; Jin et al., 2025; Dong et al., 2025; Tao et al., 2025; Li et al., 2025c) use PPO, GRPO, or DPO to optimise reasoning while adhering to syntax constraints learned during SFT. As noted in the 'Reward' column, most systems in this category rely on rule-based outcome metrics, though WebThinker uniquely employs iterative preference optimisation to align dynamically with expert demonstrations.

The second is the **Direct Paradigm** (*Direct*), where agents bypass SFT or employ specialised objectives directly on base models. DeepResearcher, ReSearch, and WebWatcher (Zheng et al., 2025; Chen et al., 2025b; Geng et al., 2025) apply GRPO to incentivise long-horizon planning without a supervised warm-up phase. Others leverage niche algorithms for specific constraints: SWIRL employs Offline-RL to learn exclusively from pre-collected logs (Goldie et al., 2025), while Kimi-Researcher utilises REINFORCE.

The third is the **Proprietary Paradigm** (*Proprietary*), encompassing industrial systems like OpenAI DR, Gemini DR, and AutoGLM (OpenAI, 2025b; Google Team, 2025; Zhipu AI, 2025). While details are typically undisclosed, AutoGLM is notable for integrating process-based rewards to evaluate intermediate steps, contrasting with the outcome-focused metrics of the Hybrid approach.

### 3.5 Non-parametric Continual Learning

DR agents depend heavily on LRMs and often utilise complex hierarchical workflows. Parameter-based learning approaches such as SFT and RL encounter significant obstacles in this context, including the need to scale model parameters, manage extensive volumes of structured experience data, and design increasingly intricate training algorithms. In contrast, non-parametric continual learning approaches offer a scalable alternative: agents refine their capabilities at runtime by optimising external memory, workflows, and tool configurations through continuous interaction with the external environment rather than by updating internal weights. This non-parametric continual learning paradigm enables efficient online adaptation with minimal data and computational overhead, making it well-suited to DR agents with complex architectures.

Non-parametric continual learning approaches, most notably case-based reasoning (CBR), are currently a mainstream method in LLM-driven agent systems. The CBR-based method enables agents to retrieve, adapt, and reuse structured problem-solving trajectories from an external case bank dynamically. Unlike traditional RAG-based methods, which rely on static databases, CBR facilitates online contextual adaptation and effective task-level generalisation. Such flexibility underscores its potential as a scalable and practical optimisation solution for DR agents with complex architecture. DS-Agent (Guo et al., 2024) is a pioneering LLM-driven agent that introduced CBR into automated data science workflows, employing approximate online retrieval from a constructed case bank. Similarly, LAM (Guo et al., 2025) applies CBR techniques to functional test generation, combining trajectory-level retrieval with LLM planning in a modular system design. Although DS-Agent itself does not include a learning phase, Agent K (Grosnit et al., 2024)advances this paradigm with dynamic external case retrieval and reuse guided by a reward-based memory policy, which exemplifies genuine self-evolution enabling continual adaptation and optimisation without updating model parameters. Focusing on DR agents, AgentRxiv (Schmidgall & Moor, 2025) further extends this paradigm by enabling autonomous research agents to collaboratively share and access a centralised repository of prior research outputs. This framework allows LLM agent laboratories to upload and retrieve reports from a shared preprint server, simulating an online-updating arXiv-like platform, which can be seen as a comprehensive case bank. Such a system empowers agents to enhance their capabilities and knowledge through contextual adaptation without modifying their model parameters.

Compared to prompt-based methods, which encode fixed demonstrations or task heuristics into static input templates, Non-parametric methods enable dynamic retrieval and adaptation of structured trajectories, thereby facilitating continual task generalisation without manual prompt engineering. Relative to RAG, which typically retrieves unstructured textual content from static corpora, CBR operates at the trajectory level and emphasises reasoning-centred memory organisation. A notable example is the Kaggle Grandmaster Agent (Grosnit et al., 2024), which demonstrates how LLMs equipped with modular reasoning components

and persistent memory can achieve expert-level structured problem solving, aligning closely with the CBR paradigm. These characteristics make CBR particularly well-suited for agents requiring procedural adaptation and context-sensitive optimisation across tasks. Beyond memory-based methods, self-evolution can also arise from dynamic infrastructure adaptation. For example, Alita (Qiu et al., 2025) monitors task requirements and environmental signals to provision and configure new MCP servers at runtime, directly extending and refining its toolset on demand.

In summary, these self-evolution paradigms in LLM-driven DR agent systems offer substantial promise for structured reasoning and dynamic retrieval and open new pathways for efficient knowledge reuse and continual learning. Although these methods have not yet achieved widespread attention, they address the high data and computational demands inherent to parameter-based approaches and therefore represent an attractive direction for future research and practical deployment.

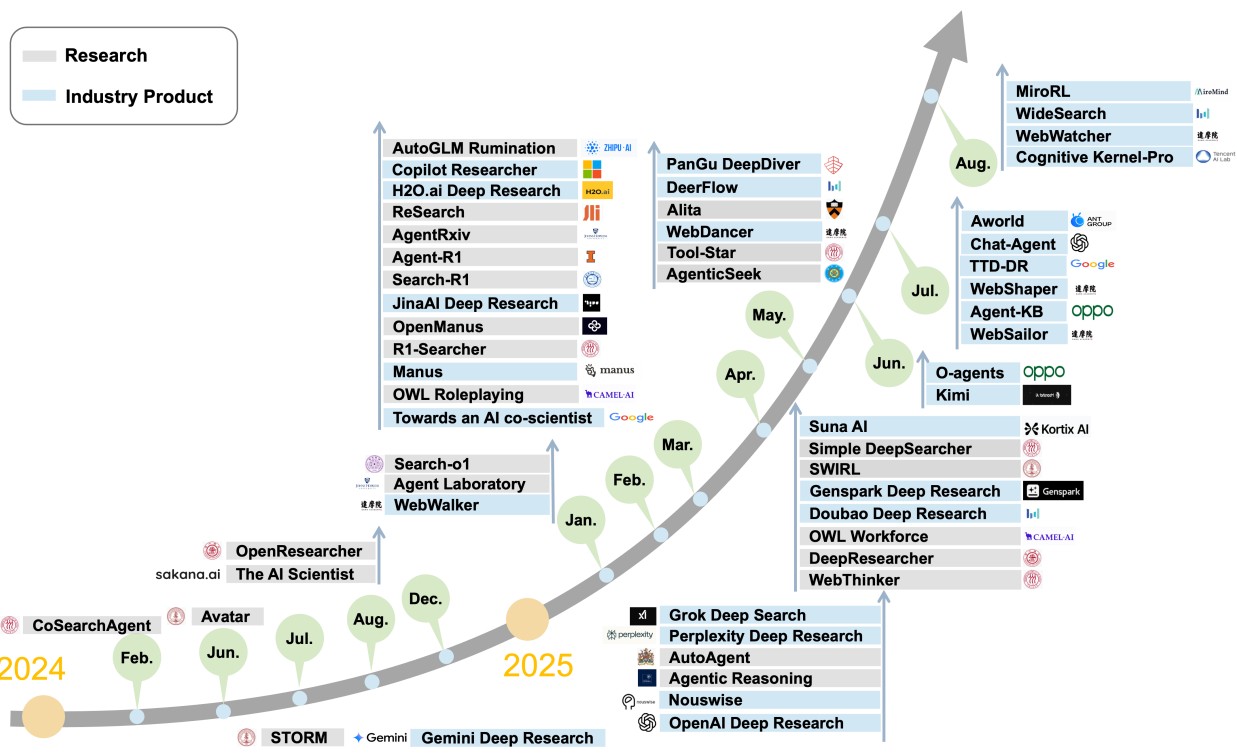

Figure 5: An overview of DR agents' evolution over years.

## 4 Industrial Applications of Deep Research Agents

The transition of Deep Research (DR) from academic prototypes to production-grade systems has precipitated a diverse landscape of industrial applications. As summarised in Fig. 5, the release cadence accelerated markedly over 2024-2025, and industrial deployments diversified beyond a single canonical architecture. Rather than a monolithic adoption of a single architecture, industrial implementations have diverged into distinct

paradigms optimised for different user needs, ranging from open-ended reasoning to enterprise-grade data analytics. In this section, we adopt a usage-centric categorisation of industrial DR agents into three groups: (i) Generalist Reasoning Engines, which prioritise long-horizon planning and autonomous tool orchestration; (ii) Search-Centric Retrieval Agents, which focus on iterative web synthesis and high-throughput processing; and (iii) Enterprise Integration Agents, which ground research in proprietary data environments under stricter governance constraints. Importantly, these usage-centric categories are not a separate taxonomy: each group can be mapped back to the workflow axes in Fig. 4 (static/dynamic, planning strategy, and single-/multi-agent topology), enabling consistent comparisons across academic and industrial systems. Table 6 in Appendix A provides a comparative overview of these systems.

## 4.1 Generalist Reasoning Engines

Agents in this category are characterised by their employ reinforcement learning (RL) to facilitate long-horizon planning, self-correction, and tool orchestration. They function as autonomous generalists capable of handling disparate tasks, from literature review to code-based data visualisation.

**OpenAI Deep Research.** As a representative example in this category, OpenAI's system (OpenAI, 2025b) is described as a single-agent architecture centred around an RL-fine-tuned reasoning model (o3). Its core innovation lies in a dynamically adaptive workflow that autonomously formulates and executes multi-step strategies. According to OpenAI (OpenAI, 2025b), the system combines web browsing with a Python sandbox to support computational steps (e.g., statistical analysis and chart generation). Public materials also state an intent to reduce unsupported claims; however, the extent of "logical verification" is not specified.

**Gemini and Qwen Deep Research.** Adopting a similar generalist scope, Google's Gemini DR (Google Team, 2025) leverages the multimodal Gemini 2.0 model. It differentiates itself through asynchronous task management, enabling the parallel execution of multiple research sub-threads, and interactive research planning, where the agent proposes structured plans for user refinement. Similarly, Alibaba's Qwen DR utilises a unified agent framework to enable concurrent task orchestration. Through RL-optimised scheduling, it enables simultaneous validation and synthesis of retrieved information, reportedly reducing latency in complex workflows, though no comparative benchmarks have been publicly released.

**AutoGLM.** Extending the generalist paradigm, Zhipu AI's AutoGLM (Zhipu AI, 2025) introduces a *Computer Use* paradigm. Unlike agents limited to API calls, it performs autonomous GUI navigation, enabling cross-application workflows and session management. It incorporates a rumination-based self-correction mechanism to validate intermediate steps, ensuring reliability in dynamic digital environments.

## 4.2 Search-Centric Retrieval Agents

In contrast to the heavy computational reasoning of generalist engines, search-centric agents prioritise the speed, breadth, and synthesis of web information. These systems often employ specialised architectures to handle high-frequency retrieval and massive token ingestion.

**Perplexity and Grok DeepSearch.** Perplexity's agent (Perplexity Team, 2025) focuses on iterative information retrieval, utilising a dynamic prompt-guided model selection mechanism to autonomously switch between specialised models based on query complexity. Similarly, xAI's Grok DeepSearch (xAI Team, 2025) introduces a segment-level processing pipeline. It employs sparse attention mechanisms to undertake

concurrent reasoning subtasks, such as data cleaning and cross-source verification in real-time, augmented by a secure sandbox for verifying computational claims and 3D visualisations.

**Kimi K2 Deep Research.** Moonshot AI's Kimi K2 (Team et al., 2025) adopts a distinct approach based on token-efficient learning and a scalable sparse architecture. By utilising a mixture-of-experts (MoE) design with multi-head latent attention, Kimi K2 manages explicit memory-aware execution on H800 clusters. Its post-training phase uniquely combines verifiable rewards for objective tasks with rubric-based self-critique for subjective inquiries, enabling it to process and synthesise information from a vetted corpus of approximately 15.5 trillion tokens. Jina AI (AI, 2025a) complements this sector by providing specialised retrieval-augmented generation (RAG) tools that convert complex web structures into LLM-optimised formats.

### 4.3 Enterprise and Analytics Integration

The third category adapts DR capabilities for closed-domain, high-security enterprise environments. These agents are distinguished not by their open-web crawling, but by their ability to interface with proprietary data lakes and internal documents.

**Microsoft Copilot Researcher and Analyst.** Microsoft's approach bifurcates the research capability into two specialised agents (Spataro, 2025). Researcher is designed for qualitative synthesis, securely accessing users' work data (emails, meeting notes) alongside web information to support research workflows such as market analysis, according to Microsoft's official announcement (Spataro, 2025). Conversely, Analyst focuses on quantitative rigour. Leveraging the o3-mini model optimised for analytical tasks, it is described as transforming raw data into insights using a chain-of-thought reasoning approach within the secure boundary of the Microsoft 365 ecosystem (Spataro, 2025). Independent benchmarks for these enterprise capabilities have not been publicly released.

**H2O.** Targeting regulated industries, H2O.ai (H2O.ai, 2025) offers an enterprise-grade solution focusing on private, on-premise deployment. It specialises in creating audit trails and managing internal document archives, prioritising data sovereignty and compliance over open-ended web exploration.

## 5 Benchmarks for DR Agent

Evaluating DR agents requires benchmarks that capture their full research workflow, including multi-step information retrieval, cross-source synthesis, dynamic tool invocation, and structured evidence-grounded report generation. Existing evaluations fall into two main categories. **Question-Answering (QA)** benchmarks range from single-turn factual queries to complex research-style problems, assessing agents' factual knowledge, domain-specific reasoning, and ability to locate and integrate relevant information. **Task Execution benchmarks** evaluate broader capabilities such as long-horizon planning, multimodal understanding, tool usage, and environment interaction by measuring how well agents carry out end-to-end research tasks. Although long-form generation datasets such as Qasper (Dasigi et al., 2021) and ELI5 (Fan et al., 2019) provide tests of extended output coherence, their free-form nature does not align with the structured evidence-based reporting expected of DR agents. Consequently, there is a pressing need for specialised benchmarks that reflect the multi-stage, multimodal characteristics of DR workflows and ensure rigorous and relevant assessment of agent performance across all phases of autonomous research.

Table 4: Performance of DR agents on major QA benchmarks. The best performance is highlighted in **bold**, and the second-best is indicated with an underline.
□ = **Not present**

| DR Agent | Base Model | QA Benchmarks | | | | | Release |
| | | Hotpot | 2Wiki | NQ | TQ | GPQA | |
| --- | --- | --- | --- | --- | --- | --- | --- |
| Search-o1 (Li et al., 2025b) | QwQ-32B-preview | 57.3 | **71.4** | 49.7 | 74.1 | 57.9 | Jan-2025 |
| Agentic Reasoning (Wu et al., 2025c) | DeepSeek-R1, Qwen2.5 | □ | □ | □ | □ | 67.0 | Feb-2025 |
| Grok DeepSearch (xAI Team, 2025) | Grok3 | □ | □ | □ | □ | **84.6** | Feb-2025 |
| AgentRxiv (Schmidgall & Moor, 2025) | GPT-4o-mini | □ | □ | □ | □ | 41.0 | Mar-2025 |
| R1-Searcher (Song et al., 2025) | Qwen2.5-7B-Base | 71.9 | 63.8 | □ | □ | □ | Mar-2025 |
| ReSearch (Chen et al., 2025b) | Qwen2.5-32B-Inst | 67.7 | 50.0 | □ | □ | □ | Mar-2025 |
| Search-R1 (Jin et al., 2025) | Qwen2.5-7B-Inst | 34.5 | 36.9 | 40.9 | 55.2 | □ | Mar-2025 |
| DeepResearcher (Zheng et al., 2025) | Qwen2.5-7B-Inst | 64.3 | 66.6 | **61.9** | **85.0** | □ | Apr-2025 |
| WebThinker (Li et al., 2025c) | QwQ-32B | □ | □ | □ | □ | 68.7 | Apr-2025 |
| SimpleDeepSearch (Sun* et al., 2025) | QwQ-32B | **73.5** | □ | □ | □ | □ | Apr-2025 |
| SWIRL (Goldie et al., 2025) | Gemma-2-27B | 72.0 | □ | □ | □ | □ | Apr-2025 |
| Tool-Star (Dong et al., 2025) | Qwen2.5-3B | 51.9 | 40.0 | □ | □ | □ | May-2025 |

**QA Benchmarks.** QA benchmarks span a spectrum of complexity, from simple factual recall to multi-hop reasoning and research-style question answering, which is provided in Table 7 in Appendix A At the lower end, datasets such as **SimpleQA** (Wei et al., 2024), **TriviaQA** (Joshi et al., 2017), and **PopQA** (Mallen et al., 2023)focus on parametric or single-hop factual recall, evaluating whether models can retrieve short factual answers from memory or minimal context. **Natural Questions (NQ)** (Kwiatkowski et al., 2019) and **TELEQnA** (Maatouk et al., 2023) add complexity by requiring answer extraction from long documents or domain-specific sources. Benchmarks like **HotpotQA** (Yang et al., 2018), **2WikiMultihopQA** (Ho et al., 2020), and **Bamboogle** (Aksitov et al., 2023) emphasize multi-hop reasoning and supporting evidence selection across documents. At the highest level of difficulty lies **Humanity's Last Exam (HLE)** (Phan et al., 2025), which targets expert-level, open-domain scientific questions crafted by leading professors in various fields. These questions often require multi-turn retrieval, complex inference, and even multimodal understanding. Additionally, BrowseComp (OpenAI Team, 2025) is another challenging benchmark proposed by OpenAI to measure the ability of AI agents to locate hard-to-find information. It retains the answer verifiability of the Simple QA benchmark while filtering out those that can be easily solved by LLMs with web search, thus testing agents' information retrieval and synthesis capabilities. Despite recent advancements, leading DR agents still exhibit suboptimal performance on the HLE and BrowserComp benchmark compared to human experts. This highlights these two benchmarks as the most critical and unresolved challenges in the evaluation of DR agents.

**Task Execution Benchmarks.** Task execution benchmarks evaluate an agent's integrated capabilities in tool use, environment perception, and information filtering. These can be grouped into two subcategories. The first category comprises general-purpose assistant tasks such as GAIA (Mialon et al., 2023), AssistantBench

Table 5: Performance of DR agents on GAIA test and validation sets. The best performance is highlighted in **bold**, and the second-best is indicated with an underline.
☐ = **Not present**

| DR Agent | Base Model | GAIA | | | | Release |
| | | Level-1 | Level-2 | Level-3 | Ave. | |
|---|---|---|---|---|---|---|
| **Test set** | | | | | | |
| MMAC-Copilot (Song et al., 2024) | GPT-3.5, GPT-4 | 45.16 | 20.75 | 6.12 | 25.91 | Mar-2024 |
| H2O.ai DR (H2O.ai, 2025) | Claude3.7-Sonnet | 89.25 | **79.87** | **61.22** | **79.73** | Mar-2025 |
| Alita (Qiu et al., 2025) | Claude-Sonnet-4, GPT-4o | **92.47** | 71.7 | 55.1 | 75.42 | May-2025 |
| Agent-KB (Tang et al., 2025b) | GPT-4.1, Claude-3.7 | 84.91 | 74.42 | 57.69 | 75.15 | Jul-2025 |
| O-agents (Zhu et al., 2025a) | Claude-3.7 | 83.02 | 74.42 | 53.85 | 73.93 | Jun-2025 |
| WebDancer (Wu et al., 2025a) | QwQ-32B | 61.5 | 50.0 | 25.0 | 51.5 | May-2025 |
| WebShaper (Tao et al., 2025) | Qwen-2.5-72B | 69.2 | 63.4 | 16.6 | 60.1 | Jul-2025 |
| Deep Researcher with Test-Time Diffusion Han et al. (2025) | Gemini-2.5-Pro | ☐ | ☐ | ☐ | 69.1 | Jul-2025 |
| Cognitive Kernel-Pro (Wan et al., 2025) | Claude-3-7 | 83.02 | 68.60 | 53.85 | 70.91 | Aug-2025 |
| **Dev set** | | | | | | |
| AutoAgent (Tang et al., 2025a) | Claude-Sonnet-3.5 | 71.7 | 53.5 | 26.9 | 55.2 | Feb-2025 |
| OpenAI DR (OpenAI, 2025b) | GPT-o3-customized | 78.7 | **73.2** | 58.0 | 67.4 | Feb-2025 |
| Manus (Manus AI, 2025) | Claude3.5, GPT-4o | 86.5 | 70.1 | 57.7 | 71.4 | Mar-2025 |
| OWL (CAMEL-AI.org, 2025) | Claude-3.7-Sonnet | 84.9 | 68.6 | 42.3 | 69.7 | Mar-2025 |
| H2O.ai DR (H2O.ai, 2025) | h2ogpt-oasst1-512-12b | 67.92 | 67.44 | 42.31 | 63.64 | Mar-2025 |
| Genspark Super Agent (Team, 2025b) | Claude 3 Opus | **87.8** | 72.7 | 58.8 | **73.1** | Apr-2025 |
| WebThinker (Li et al., 2025c) | QwQ-32B | 53.8 | 44.2 | 16.7 | 44.7 | Apr-2025 |
| SimpleDeepSearch (Sun* et al., 2025) | QwQ-32B | 50.5 | 45.8 | 13.8 | 43.9 | Apr-2025 |
| Alita (Qiu et al., 2025) | Claude-Sonnet-4, GPT-4o | 75.15 | ☐ | **87.27** | ☐ | May-2025 |

(Yoran et al., 2024), and Magentic-One (Fourney et al., 2024). These tasks require agents to plan and execute tool-based workflows (for example, searching, browsing, or form filling) within environments that are open-ended and often web-based. Among them, **GAIA** has emerged as the most important benchmark, offering diverse, realistic tasks that are easily human-solvable but remain highly challenging for current agents. The second subcategory focuses on **research and code-oriented tasks**, including **SWE-bench** (Jimenez et al., 2024), **HumanEvalFix** (Muennighoff et al., 2024), **MLGym** (Nathani et al., 2025), **MLE-bench** (Chan & Others, 2025), **MLBench** (Tang et al., 2024), **MLAgentBench** (Huang et al., 2024), and **ScienceAgentBench** (Chen et al., 2025d), which test agents on completing machine learning pipelines, repairing real-world code, or replicating scientific experiments. These tasks require long-horizon planning, precise tool invocation, and often code generation and validation. Additionally, benchmarks like **RE-Bench** (Wijk et al., 2024) and **RESEARCHTOWN** (Yu et al., 2024a) simulate multi-agent research environments, evaluating how well agents collaborate and iterate in multi-role scientific workflows.

As DR agents continue to integrate more interactive tools, future evaluation may expand into GUI-based manipulation environments. Benchmarks such as **OSWorld** (Xie et al., 2024), **WebArena** (Zhou et al., 2024), and **SpaBench** (Chen et al., 2025a) allow agents to control applications or web interfaces directly, opening new avenues for testing embodied research capabilities in realistic, user-facing scenarios.

## 6 Challenges and Future Directions

Despite the demonstrated efficacy of DR agents in automating multi-step information discovery (Section 3), two overarching limitations define the roadmap for future innovation. First, information acquisition remains constrained by reliance on static repositories and public web searches, overlooking the vast deep web of proprietary data. Second, system architectures are often monolithic and linear, struggling with the high-throughput, parallel reasoning required for complex inquiry. Addressing these challenges is critical to evolving DR agents from passive tools into autonomous research partners.

### 6.1 Expanding Information Horizons

As detailed in the analysis of search modalities in Section 3.1, most current agents rely on standard commercial search APIs or human-centric browsers. This architectural choice creates limitations regarding gated content and introduces execution fragility.

**Accessing Proprietary and Gated Data.** Current agents are often blind to information concealed behind proprietary interfaces, such as enterprise software or specialised financial terminals. To surmount this, future frameworks must integrate a granular range of modular tools via protocols like the MCP. This approach enables agents to dynamically access specialised APIs and databases beyond the scope of standard browsers, delivering precise, context-aware market intelligence that standard crawling cannot reach.

**Security and Governance Risks.** Expanding information access through MCP and tool integration introduces security challenges that current DR architectures have not adequately addressed. Prompt injection attacks, where malicious content embedded in retrieved documents manipulates agent behaviour, pose a risk when agents process untrusted web content. Similarly, tool injection through compromised MCP servers may return adversarial inputs that corrupt reasoning chains. Furthermore, accessing proprietary data necessitates robust authentication, fine-grained authorization, and comprehensive audit logging to comply with data

governance requirements. Addressing these security concerns is a prerequisite for the practical deployment of DR agents in enterprise and sensitive domains.

**Next-Generation Agentic Browsers.** Even when data is accessible, conventional human-centred browsers create bottlenecks due to visual rendering overhead and anti-bot defences. As noted in the discussion of industrial agents (Section 4), systems like Grok and AutoGLM are beginning to tackle this. The future lies in AI-native browsers, such as Browserbase (Adam et al., 2024), Browser Use (Müller & Žunič, 2024), Dia, Fellou (Team, 2025a), and the Comet (Team, 2024), which expose a stable, structured DOM view for programmatic traversal. By executing pages asynchronously in headless containers and embedding vision-language models to resolve login gates (Adam et al., 2024), these tools eliminate the fragility of coordinate-based actions, enabling stable, high-concurrency data harvesting at scale.

## 6.2 Robustness in Reasoning and Architecture

Section 3.4 highlighted the shift from prompting to parametric optimisation. However, achieving human-level reliability requires further advancements in verification loops and architectural parallelism.

**Verification and Self-Reflection.** To mitigate hallucinations, future agents must move beyond simple generation to structured verification loops. Emerging strategies involve proactive cross-checks: rating source credibility, inspecting consistency across multiple layers, and verifying claims against independent origins, as exemplified by Grok DeepSearch (xAI Team, 2025). Furthermore, incorporating *meta-cognition* or *rumination*, where agents pause to inspect intermediate results before finalising answers (Zhipu AI, 2025), will be essential. By adding correctness-oriented rewards in RL, agents can learn to backtrack and revise earlier inferences when conflicts arise (OpenAI, 2025b).

**Adaptive Reasoning via Tool Use.** A fundamental challenge lies in extending simple tool usage to complex, multi-step reasoning. Traditional supervised fine-tuning often leads to over-reasoning or inappropriate tool selection. Recent work by Qian et al. (2025) demonstrates that RL frameworks with fine-grained rewards, evaluating not just the final answer but also tool selection appropriateness and parameter accuracy, can significantly enhance performance. Future DR agents must adopt this integrated approach to achieve superior generalisation to unseen tools and more rational invocation patterns.

**Scalable Multi-Agent Architectures.** While single-agent models (e.g., OpenAI o3 and Agent-R1 (OpenAI, 2025b; Ouyang et al., 2025)) demonstrate strong reasoning, they face cognitive bottlenecks when simultaneously managing planning, execution, and reporting. Reflecting the coordination challenges identified in Section 3.3.1 (Dynamic Workflows), distributing workloads across multiple specialised agents has shown promising improvements (Wang et al., 2024a), yet the credit assignment problem remains a hurdle. The future lies in parametric optimisation of multi-agent architectures. Two promising directions include: (i) Adopting Hierarchical Reinforcement Learning (HRL) with layered internal rewards to foster cooperative learning; and (ii) Employing RL-based scheduling agents that dynamically allocate subtasks based on runtime signals. This shift from linear planning to asynchronous, DAG-based execution is essential for handling high-throughput research workflows. However, asynchronous parallel execution introduces specific failure modes that remain under-explored. These include result deduplication when multiple agents retrieve overlapping information from different sources, shared-memory consistency when concurrent agents update a common knowledge store, and conflict resolution when parallel reasoning branches yield contradictory conclusions. Future benchmarks should therefore incorporate cost-aware and process-aware metrics, such as redundancy ratio, inter-agent

conflict rate, and computational cost per resolved query to evaluate not only final output quality but also the efficiency of the collaborative process.

## 6.3 Evolution and Evaluation

Building on the non-parametric continual learning paradigm discussed in Section 3.5, current self-evolution mechanisms remain underdeveloped. Finally, the sustainability of DR agents depends on their ability to self-improve and the validity of the benchmarks used to assess them.

**Continuous Self-Evolution.** Current self-evolution remains underdeveloped. Future research should expand into (i) Comprehensive Case-Based Reasoning, leveraging hierarchical experience traces (as proposed in foundational works like Aamodt & Plaza, 1994) for context-specific adaptation, and (ii) Autonomous Workflow Evolution, where agent workflows are represented as mutable structures refined via evolutionary algorithms. Systems like CycleResearcher (Weng et al., 2024) enable full process simulation through iterative preference learning with robust verifiers (Zhu et al., 2025b), sharing conceptual similarities with self-evolving paradigms like AlphaEvolve (Novikov et al., 2025). AgentRxiv (Schmidgall & Moor, 2025) complements this by facilitating the online sharing and reuse of structured research experiences.

**Dynamic and Realistic Evaluation.** Our review of existing benchmarks in Section 5 reveals a critical disconnect between current static datasets and genuine research capabilities. Most evaluations rely on QA suites (e.g., Wikipedia-based) that are easily hacked by parametric memory. The field urgently needs Open-Web, Time-Sensitive Benchmarks like BrowseComp (OpenAI Team, 2025) that force agents to locate hard-to-find, real-time information. Furthermore, metrics must evolve beyond simple fact retrieval to assess End-to-End Report Generation, evaluating the agent's ability to weave textual narrative, tables, and citations into a coherent, multimodal discourse.

## 6.4 Ethical Considerations and Broader Impact

While Deep Research agents significantly enhance productivity, they introduce critical ethical concerns that warrant attention alongside technical advancements.

**Academic Integrity and Hallucination.** A primary risk is the generation of persuasive but hallucinated citations. Unlike standard search engines that return direct links, DR agents synthesize information, potentially conflating sources or fabricating data points in an effort to satisfy the user's request for a comprehensive report. This necessitates the development of rigorous citation verification protocols within the generation loop.

**Information Echo Chambers.** DR agents that rely heavily on SEO-optimized public web searches may inadvertently reinforce bias. By prioritizing accessible, high-ranking content over less indexed but potentially more authoritative niche sources, these agents risk narrowing the diversity of research perspectives. Future research must address how to incentivize the retrieval of long-horizon knowledge.

**Proliferation of Low-Quality Content.** The ability to autonomously generate long-form, structured reports at scale lowers the barrier for creating academic-sounding spam. This poses a challenge for the scientific community in distinguishing between human-curated insights and machine-generated aggregations, underscoring the need for provenance tracking and watermarking in agent-generated outputs.

## 7  Conclusion

LLM-driven Deep Research Agents represent an emerging paradigm for automated research support, integrating advanced techniques such as iterative information retrieval, long-form content generation, autonomous planning, and sophisticated tool utilisation. In this survey, we systematically reviewed recent advancements in DR agents, categorising existing methodologies into prompt-based, fine-tuning-based, and reinforcement learning-based approaches from the perspectives of information retrieval and report generation. Non-parametric methods utilise LLMs and carefully designed prompts to achieve efficient and cost-effective deployment, making them suitable for rapid prototyping. In contrast, fine-tuning and reinforcement learning approaches explicitly optimise model parameters, and can enhance agents' reasoning and decision-making capabilities in reported evaluations. We also examined prominent DR agent systems developed by industry leaders and discussed their technical implementations, strengths, and limitations.

### Limitation

Despite notable progress, key challenges remain, including limited generalisation across diverse tasks, inflexible task workflows, difficulty in integrating granular external tools, and substantial computational complexity associated with advanced planning and optimisation. Future research directions thus emphasise broader and more flexible tool integration through modular capability providers (e.g., Operator-based architectures), development of asynchronous and parallel planning frameworks (e.g., Directed Acyclic Graph-based approaches), and sophisticated end-to-end optimisation methods for multi-agent architectures, such as hierarchical reinforcement learning or multi-stage fine-tuning pipelines. With continued advancements in LLM technologies, DR agents show promise for supporting complex research workflows and augmenting human productivity across academic and industrial domains.

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

# A  Supplementary Data and Tables

This appendix provides supporting data for the main analysis. **Table 6** presents a detailed taxonomy of industrial Deep Research agents, categorizing them by functional focus. **Table 7** provides the specific metadata (size, domain, reasoning depth) for the QA benchmark datasets discussed in the evaluation section.

Table 6: Taxonomy of Industrial Deep Research Agents. Systems are categorised by their primary functional focus: **Generalist** (Broad reasoning & planning), **Search-Centric** (High-speed retrieval & synthesis), or **Enterprise** (Internal data & analytics).

| Agent | Provider | Category | Key Technical Distinctions |
|---|---|---|---|
| **OpenAI DR** | OpenAI | Generalist | RL-optimised single-agent; dynamic workflow refinement; integrated Python environment for data visualisation. |
| **Gemini DR** | Google | Generalist | Asynchronous task management; large context RAG ensembles; interactive plan formulation. |
| **Qwen DR** | Alibaba | Generalist | Unified agent framework with RL-optimised scheduling; concurrent retrieval validation. |
| **AutoGLM** | Zhipu AI | Generalist | "Computer Use" capability; autonomous GUI navigation for cross-app workflows; rumination-based self-correction. |
| **Perplexity DR** | Perplexity | Search-Centric | Dynamic model selection; iterative query decomposition; emphasis on source authority. |
| **Grok DeepSearch** | xAI | Search-Centric | Real-time segment-level processing; sparse attention for multimodal integration; secure sandbox. |
| **Kimi K2** | Moonshot | Search-Centric | Token-efficient pretraining; sparse MoE architecture; rubric-based self-critique for subjective tasks. |
| **Jina Reader** | Jina AI | Search-Centric | Optimised for converting complex web content into LLM-friendly formats; high-performance reranking. |
| **Copilot Researcher** | Microsoft | Enterprise | Integration with M365 graph (emails, chats); secure access to third-party connectors (Salesforce). |
| **Copilot Analyst** | Microsoft | Enterprise | Specialised o3-mini inference for data analytics; chain-of-thought transformation of raw data. |
| **H2O GPTe** | H2O.ai | Enterprise | Private on-premise deployment; specialised RAG for internal document archives; auditability. |

Table 7: Overview of nine widely used QA benchmark datasets employed in recent DR-agent studies. The first group covers single-hop QA tasks, while the second group focuses on multi-hop and multi-turn reasoning.

| Benchmark | Release | Size | Task & Context | Domain | Multi-hop Nums |
|---|---|---|---|---|---|
| TriviaQA (Joshi et al., 2017) | 2017 | 95 k | Single-hop retrieval (Long web/Wiki docs) | Open | 1 |
| Natural Questions (Kwiatkowski et al., 2019) | 2019 | 307 k | Document answer extraction (Full Wikipedia article) | Open | 1 |
| PopQA (Mallen et al., 2023) | 2023 | 14 k | Single-hop parametric recall (None) | Open | 1 |
| TELEQnA (Maatouk et al., 2023) | 2023 | 10 k | Domain factual QA (Telecom standards/articles) | Telecom | 1 |
| SimpleQA (Wei et al., 2024) | 2024 | 4.3 k | Single-hop factual recall (None / parametric) | Open | 1 |
| HotpotQA (Yang et al., 2018) | 2018 | 113 k | Multi-hop reasoning (2 Wikipedia paragraphs) | Open | 2 |
| 2WikiMultihopQA (Ho et al., 2020) | 2020 | 192 k | Multi-hop reasoning (Retrieval across Wikipedia) | Open | 2+ |
| Bamboogle (Aksitov et al., 2023) | 2023 | 125 | Compositional reasoning (Online search) | Open | 2–3 |
| Humanity's Last Exam (Phan et al., 2025) | 2025 | 2.5 k | Expert-level multi-turn (Mixed external sources) | Multi-discipline | 2+ |

