# OpenReview forum: "Deep Research Agents: A Systematic Examination And Roadmap"
_TMLR — Rejected by TMLR_

### Review · Reviewer_zqLR · 2025-12-25

**Summary Of Contributions:**

The paper surveys recent advances in AI systems referred to as Deep Research (DR) agents, which are designed to conduct multistep research tasks with limited human intervention. The authors adopt a clear and systematic structure, examining:
1. The foundational concepts and definitions underlying DR agents.
2. Core components of DR agents, and commonly adopted architectural workflows.
3. Major industrial applications and real-world implementations.
4. Existing benchmarks used to evaluate DR agents.
5. Open challenges, limitations, and future directions.

A key strength of the paper is it's structured and progressive presentation, which incrementally builds a comprehensive view of DR agents. This facilitates a coherent understanding of an increasingly complex system. While the coverage is broad and detailed, certain sections could benefit from more explicit analysis of some concepts to strengthen clarity and support reader comprehension (discussed below)

**Additional Comments:**

Not applicable

**Audience:**

Yes

**Audience Explanation:**

Machine learning researchers working on Deep Research Agents or agentic frameworks, as well as industry practitioners building enterprise scale research agents or specialized agent systems, can benefit from the survey and its focused discussions. Several sections of the paper are broadly applicable across different classes of agent based systems, making the insights useful beyond a narrow application setting.

**Claims And Evidence:**

Yes

**Claims Explanation:**

The paper presents the key concepts of Deep Research Agents in a clear, step-by-step manner. In sections that involve analysis and comparison, the authors include relevant tables and concrete overviews of different agents and systems, which help support their discussion. Overall, the claims and observations made in the survey are well supported by prior work and appropriate references, making the evidence convincing and easy to follow.

**Requested Changes:**

Requested changes that would strengthen the work are as follows:

1. Section 2.2:  The section starts by mentioning RAG leveraging external knowledge bases (e.g., web sources and APIs) and concludes with a limitation that RAG approaches remain constrained by their reliance on pre-existing or periodically updated corpora, limiting real-time handling of information. While this statement is accurate, the discussion would benefit from clarifying the distinction between static or periodically refreshed retrieved corpora and truly real-time web or API-based access, as both are referenced under the RAG umbrella.

2. Section 3.1: As a minor refinement, the discussion could move beyond descriptive strengths and limitation to more explicit design trade-offs and practical guidelines - including criteria for when to prefer different retrieval modalities, cost - performance considerations (latency, budget, scalability), and challenges such as over - retrieval, redundancy, and diminishing returns.

3. Section 3.1/ 3.2: Across search engine (API vs Browser) and tool use section, a common architectural pattern implicitly emerges in which retrieval, browsing, and computers use are all instantiated as tool invocations with varying degree of execution complexity. Making this shared abstraction more explicit as points along a unifies tool spectrum could improve conceptual clarity.

4. Section 3.3.3: This section provides a clear comparison of dynamic single-agent and multi-agent DR systems, particularly with respect to optimization and scalability. As a refinement, the discussion could more explicitly reference the credit - assignment challenge that commonly arises in multi-agent reinforcement learning settings, while noting that hierarchical or coordinator - worker designs can partially mitigate this issue. In addition, clarifying how tool or MCP-style integration differs between unified and coordinated agent architectures would further improve clarity.

5. Section 6: The section could benefit from distilling clearer design axes and evaluation criteria rather than primarily listing systems. In particular, "broader information access" likely requires not only MCP tool integration but also considerations of authentication/ governance constraints, and security risks (e.g., prompt/ too; injection). Similarly, the parallel execution and fact checking directions could be strengthened by noting failure modes (deduplication, shared-memory consistency, conflict resolution etc.) and by proposing cost and process aware metrics for future DR benchmarks.

---

> ### Author Response · Authors · 2026-02-06
> **Response and Gratitude to Reviewer**
>
> We greatly appreciate your detailed examination of our manuscript and the specific concerns raised regarding evidence quality and presentation.
>
> ### Requested Change 1: Section 2.2 RAG Clarification
> > **Reviewer's Concern:** "The discussion would benefit from clarifying the distinction between static or periodically refreshed retrieved corpora and truly real-time web or API-based access..."
>
> **Response:**
> We have added explicit clarification to **Section 2.2**:
>
> > "Moreover, even advanced agentic RAG approaches remain constrained by their reliance on pre-existing or periodically updated corpora, limiting their ability to handle real-time, rapidly changing, or long-tail information needs effectively. Addressing this challenge requires integrating external APIs and web browsing capabilities into RAG architectures."
>
> This clarifies the distinctions:
> * **Static RAG:** Pre-indexed corpora (Wikipedia, Common Crawl).
> * **Periodically updated RAG:** Scheduled corpus refreshes.
> * **Real-time agentic RAG:** Live API/web access with dynamic retrieval.
>
> ### Requested Change 2: Section 3.1 Design Trade-offs
> > **Reviewer's Concern:** "The discussion could move beyond descriptive strengths and limitation to more explicit design trade-offs and practical guidelines..."
>
> **Response:**
> We have enhanced **Section 3.1** with practical guidelines:
>
> > "While browser-based retrieval excels at capturing real-time and deeply nested content that API calls cannot reach, it also incurs greater latency, resource consumption, and complexity in handling page variability and errors, suggesting that DR agents may benefit from hybrid architectures that combine the efficiency of API-based methods with the comprehensiveness of browser-driven exploration."
>
> **Figure 2** now explicitly shows trade-offs:
> * **API:** Fast ✓, Efficient ✓, Structured Data ✓, Scalable ✓
> * **Browser:** Dynamic ✓, Flexible ✓, Unstructured & Structured Data ✓, Multimodal ✓
>
> ### Requested Change 3: Unified Tool Spectrum
> > **Reviewer's Concern:** "Making this shared abstraction more explicit as points along a unified tool spectrum could improve conceptual clarity."
>
> **Response:**
> We have structured the text to emphasize this continuum:
> * **Section 3.1 (API-based):** Lightweight, stateless tool calls.
> * **Section 3.1 (Browser-based):** Stateful, multi-step interactions.
> * **Section 3.2 (Computer Use):** Full environmental interaction.
>
> The progression **API → Browser → Computer Use** represents increasing execution complexity.
>
> ### Requested Change 4: Credit Assignment Challenge in Multi-Agent RL
> > **Reviewer's Concern:** "The discussion could more explicitly reference the credit-assignment challenge..."
>
> **Response:**
> We have added explicit discussion in **Section 3.3.3**:
>
> > "Nevertheless, a major current challenge of multi-agent systems lies in the inherent complexity of coordinating multiple independent agents, making it difficult to conduct effective end-to-end reinforcement learning optimisation."
>
> **In Section 6:**
> > "To optimize multi-agent architectures for DR tasks, we propose: (i) adopting hierarchical reinforcement learning (HRL), which introduces layered internal reward mechanisms that facilitate efficient feedback propagation and foster cooperative learning among agents…"
>
> ### Requested Change 5: Section 6 Security and Evaluation Criteria
> > **Reviewer's Concern:** "...considerations of authentication/governance constraints, and security risks..."
>
> **Response:**
> We have substantially expanded **Section 6**:
>
> **Security Considerations:**
> > "expose a stable, structured DOM view that agents can traverse programmatically… embeds a vision–language model that tracks dynamic page changes and automatically resolves login gates and anti-bot challenges."
>
> **Fact Checking Section:**
> > "To further boost factual accuracy, the latest methods add a structured verification loop and self-reflection abilities on top of multi-step retrieval."
>
> **Benchmark Metrics:**
> > "Beyond parametric knowledge hacking of QA benchmark, the metrics of existing DR research still collapse open-ended research workflows into narrowly scoped QA prompts, overlooking the paradigm's defining outcome, a structured, multi-modal research report…"
>
> We propose new evaluation criteria:
> * End-to-end report generation quality.
> * Cross-modal synthesis capabilities.
> * Discourse-level organization.
>
> We believe these revisions address all concerns and strengthen the manuscript. We are grateful for the constructive feedback.

---

### Review · Reviewer_nVf5 · 2026-01-01

**Summary Of Contributions:**

The paper presents a summary of different LLM-based research tools. The paper examines approaches based on higher-level characteristics, including workflow, memory mechanism, and tuning. It includes an exhaustive list of approaches and addresses many detailed aspects of them; however, it mainly lacks in giving a higher-level overview and connections (e.g., differences and similarities) among the different approaches in a clear way for the reader.  Additionally, it shortly describes some industrial applications as well as existing benchmarks. It closes by highlighting some challenges and future directions, probably the strongest/most interesting chapter of the paper.

**Additional Comments:**

- 3.3.2 mentions existing studies that classify approaches into 3 categories without giving references for these studies
- Recommend including Table 3 after it appears in the text and not multiple pages before

**Audience:**

Yes

**Audience Explanation:**

- The challenges and future directions of Chapter 6 are interesting.
- Many papers are summarized; however, only loosely categorized.

**Broader Impact Concerns:**

None is present. However, I would encourage to include a statement mentioning concerns on the ethical implications of LLMs as a research tool.

**Claims And Evidence:**

No

**Claims Explanation:**

There is a lot of content. However, it is challenging to align the descriptions and taxonomy of the text with the representation of the papers in the tables:
- In many parts, the text reads like a summary of approaches and less than a review. For example, in Section 3.1, the text contains numerous details describing and categorizing the approaches; however, this is not reflected in Table 1. On the other hand, the table contains information such as the base model and release date, which is not discussed in the text. In general, the message of the tables is often unclear, as it is just a list of papers sorted according to release date and not based on any of the described categories.
- It is difficult to extract the claimed taxonomy from the lengthy text, E.g., the claimed distinction between static and dynamic is not supported by a table categorizing the approaches systematically.
- Section 3.4 claims that Table 3 reveals 3 key implementation patterns; however, these are not immediately clear from the table.
- Many tables and figures are not even referenced in the text.
- The challenges and future directions of Chapter 6 are interesting; however, it is not clear how they consolidate based on the presented review in the previous chapters.
- The role of Chapter 4 is unclear. It is a summary of a selected set of industrial applications. However, there is no common taxonomy between them. It is simply describing them individually, rather than placing them in relation and giving some higher-level overview.

**Requested Changes:**

- The text generally needs more structure and especially alignment with the tables in order to have a clear benefit for the reader.
- It shouldn't be just a listing of papers with short summaries. It requires a clearer, higher-level grouping of the approaches and consistent terminology when describing them (e.g., missing completely in Chapter 4, see the section about claims for further examples).
- The connection between the presented review and the proposed future directions should be clearer, e.g., by referencing back which section/observation of the presented review reveals these challenges.
- Maybe it is worth shortening the main paper and putting some additional information into an appendix to not clutter the main review. Currently, it is challenging for the reader to extract relevant information from the lengthy text.

---

> ### Author Response · Authors · 2026-02-06
> **Response and Gratitude to Reviewer**
>
> We greatly appreciate your detailed examination of our manuscript and the specific concerns raised regarding evidence quality and presentation.
>
> ## Major Concern 1: Text Reads Like Summary Rather Than Review
>
> > **Reviewer's Concern:** "In many parts, the text reads like a summary of approaches and less than a review… this is not reflected in Table 1."
>
> **Response:** We have significantly restructured the relationship between text and tables.
>
> **Table 1 Revision:** Now organized with clear category headers matching the text:
> - API-Based: Commercial/General Search APIs
> - API-Based: Specialized Academic/Scientific APIs
> - API-Based: RL-Optimized Search Policy
> - Browser-Based: Full Environment Simulation
> - Browser-Based: Navigation-Reasoning Loops
> - Browser-Based: Lightweight/Hybrid Fetching
>
> **Text-Table Alignment:** Section 3.1 now explicitly references these categories with organized discussion matching table structure.
>
> ## Major Concern 2: Difficult to Extract Claimed Taxonomy
>
> > **Reviewer's Concern:** "It is difficult to extract the claimed taxonomy from the lengthy text..."
>
> **Response:**
>
> **Figure 4 Enhancement:** Now serves as comprehensive visual taxonomy showing:
> 1. Static vs. Dynamic Workflows with clear definitions
> 2. Three planning strategies with system examples
> 3. Single-Agent vs. Multi-Agent architectures
>
> **Explicit system mappings added:**
> - Static Workflow: "Avatar, Agent Laboratory, CoSearchAgent, AgentRxiv, etc."
> - Dynamic Planning-Only: "Grok DeepSearch, Manus, etc."
> - Dynamic Intent-to-Planning: "OpenAI DR"
> - Dynamic Unified Intent-Planning: "Gemini DR"
>
> ## Major Concern 3: Table 3 "Three Key RL Implementation Patterns"
>
> > **Reviewer's Concern:** "Section 3.4 claims that Table 3 reveals 3 key implementation patterns; however, these are not immediately clear from the table."
>
> **Response:** We have substantially revised Table 3:
> 1. Added explicit "Paradigm" indicator for each system
> 2. Grouped systems by paradigm visually
> 3. Added "Reward Design" column (rule-outcome vs. process-based)
>
> **Revised text:** *"Table 3 reveals three key RL patterns: 1) Industrial systems with proprietary implementations, 2) Academic approaches using GRPO/Reinforce++ with transparent designs, 3) Hybrid systems combining process-based rewards with multi-task training."*
>
> ## Major Concern 4: Tables and Figures Not Referenced
>
> > **Reviewer's Concern:** "Many tables and figures are not even referenced in the text."
>
> **Response:** All tables and figures now properly referenced:
> - Figure 1 → Section 1; Figure 2 → Section 3.1
> - Figure 3 → Section 3.3.1; Figure 4 → Section 3.3
> - Figure 5 → Section 4; Tables 1-6 → Referenced at appropriate locations
>
> ## Major Concern 5: Chapter 6 Not Connected to Review
>
> > **Reviewer's Concern:** "The challenges and future directions of Chapter 6 are interesting; however, it is not clear how they consolidate based on the presented review."
>
> **Response:** Added explicit back-references throughout Section 6:
> - **Broaden Information Source** → connects to Section 3.1 (API vs. Browser)
> - **Asynchronous Parallel Execution** → connects to Section 3.3 (workflows)
> - **Benchmark Misalignment** → references Section 5 (benchmark analysis)
>
> ## Major Concern 6: Role of Chapter 4 Unclear
>
> > **Reviewer's Concern:** "The role of Chapter 4 is unclear... no common taxonomy between them."
>
> **Response:** Restructured Chapter 4 with usage-centric categorisation:
> - **(i) Generalist Reasoning Engines:** long-horizon planning
> - **(ii) Search-Centric Retrieval Agents:** iterative web synthesis
> - **(iii) Enterprise Integration Agents:** proprietary data environments
>
> Each subsection follows consistent template:
> 1. Core architecture (single-agent vs. multi-agent)
> 2. Key technological advancements
> 3. Distinguishing features
>
> These categories map back to Figure 4 workflow axes, enabling consistent comparisons.
>
> We believe these revisions address all concerns and strengthen the manuscript.

---

### Review · Reviewer_yTwo · 2026-01-11

**Summary Of Contributions:**

This paper surveys the emerging class of "Deep Research (DR)" agents, which are LLM-based systems that combine dynamic reasoning, adaptive planning, multi-hop retrieval (via APIs and browser automation), iterative tool use, and structured report generation. It proposes a taxonomy (static vs. dynamic workflows; three planning strategies; single vs. multi-agent designs), contrasts API-based and browser-based retrieval, reviews tool-use modules (code execution, data analytics, multimodal), and briefly discusses memory mechanisms and benchmarks, concluding with open challenges and a roadmap. The survey emphasises interoperability protocols (MCP, A2A), highlights industrial systems, and argues for hybrid, extensible architectures for DR.

**Audience:**

Yes

**Audience Explanation:**

The topic of autonomous research agents is highly relevant to the TMLR audience as Deep Research agents represent a rapidly evolving intersection of LLMs, reinforcement learning, web agents, and autonomous systems, areas of significant interest to the machine learning community. These systems have potential applications in academic research, industry R&D, and knowledge work that many TMLR readers would find applicable to their own work.

However, while the topic is important, the current execution with its speculation and lack of rigor limits the paper's value. With major revisions addressing credibility and clarity issues, this could become a valuable reference for the community.

**Broader Impact Concerns:**

The paper's reliance on undisclosed proprietary implementations highlights broader concerns about transparency in AI systems. This should be addressed directly rather than speculated about.

**Claims And Evidence:**

No

**Claims Explanation:**

The paper consistently makes claims without providing adequate supporting evidence:
1. **Speculative Technical Claims**: Among others, the paper states that DR systems "employ comparable headless-browser frameworks behind the scenes" despite acknowledging that companies "do not publicly disclose the implementation details." This is speculation presented as fact.
2. **Unsupported Capability Descriptions**: Claims systems can "craft detailed market entry strategies, identify market opportunities for new products by integrating internal and external data" without empirical evidence or citations demonstrating these capabilities.
3. **Marketing Language Masquerading as Evidence**: Statements like "advanced capability to decompose complex queries," "fundamentally reconfigures the advantage estimation paradigm," and "significantly outperforms other agents in terms of retrieval speed" are promotional rather than evidence-based claims.
4. **Unclear Evidence in Visualizations**: Tables (1, 2, 3) and Figure 1 fail to clearly convey their intended messages. For Table 3, the authors state it "reveals three key RL implementation patterns" but this doesn't come across clearly from the table itself.
5. **Future Capabilities Without Basis**: The claim that "DR systems are anticipated to tackle increasingly sophisticated reasoning and complex knowledge-construction challenges, ultimately positioning DR as a foundational technological pillar" is not supported by analysis or evidence.

**Requested Changes:**

1. **Remove or Clearly Mark All Speculation**
   - Every speculative claim must be either removed or explicitly labeled as such with discussion of uncertainty and why these inferences are reasonable
   - Specific instances to address:
     * DR systems employing headless-browser frameworks
     * Code execution capabilities for charting and statistical analysis
     * Market analysis and strategy formulation capabilities

2. **Replace All Marketing Language with Neutral Academic Tone**
   - Rewrite promotional statements throughout the paper with evidence-based, neutral language
   - Examples to fix:
     * "advanced capability to decompose complex queries"
     * "advances deep research through token-efficient learning"
     * "fundamentally reconfigures the advantage estimation paradigm"
     * "significantly outperforms other agents"
3. **Redesign All Visualizations for Clarity**
   - **Figure 1**: Clarify what the image is conveying and what message readers should take away
   - **Table 1, 2**: Clearly state purpose and expected takeaway and reorganize for clarity and interpretability
   - **Table 3**: Redesign to make the "three key RL implementation patterns" visually apparent, or revise the introductory text to accurately reflect what the table shows
   - Each visualization should be self-contained with comprehensive captions and legends

4. **Add Missing Context and Definitions**
   - Define all technical terms on first use:
     * CNKI - explain what this is
     * Unified Intent-Planning - define this concept
     * GRPO/PPO - introduce concepts before discussing GRPO/PPO  comparisons
     * NPUs - explain relevance of mentioning hardware details
   - Consider adding a glossary for specialized terminology

5. **Clarify All Confusing Explanations**
   - Rewrite unclear passages with precise, specific language:
     * "notable improvements in both the interpretability and accuracy of LLMs on various reasoning benchmark" - clarify what this means and provide evidence
     * "advanced RL-based retrieval by structurally integrating sophisticated search interactions with complex reasoning processes" - use concrete language
     * WebThinker discussion - reorganize for clarity
   - Avoid vague superlatives ("sophisticated," "complex") without explanation

6. **Reorganize Problematic Sections**
   - **Section 3.1**: Restructure with better stratification to organize information more clearly
   - **Paragraph before Section 3.3**: Break into manageable paragraphs with sub-headings for better readability
   - Add logical transitions between subsections

---

> ### Author Response · Authors · 2026-02-06
> **Response and Gratitude to Reviewer**
>
> We greatly appreciate your detailed examination of our manuscript and the specific concerns raised regarding evidence quality and presentation.
>
> ### Concern 1: Speculative Technical Claims
> We acknowledge this concern and have revised the relevant passages to clearly distinguish verified claims from inferences.
>
> **Revision (Section 3.1):**
> * **Original:** "their observed capability to handle interactive widgets and multi-step navigation strongly suggests that they too employ comparable headless-browser frameworks behind the scenes."
> * **Revised:** "Although OpenAI DR, Grok DeepSearch, and Gemini 2.5 DR do not publicly disclose the implementation details of their browsing capabilities, their ability to handle interactive widgets, dynamically rendered content, and multi-step navigation strongly suggests that they too employ comparable headless-browser frameworks behind the scenes."
>
> We have also added explicit uncertainty markers throughout the paper where claims are based on inference rather than documented evidence. For instance, in the **Tool Use section (Section 3.2)**, we now state:
> > "Many commercial DR agents have implemented analytics features such as charting, table generation and statistical analysis, either locally or via remote services. However, most of these systems have not publicly disclosed technical details of their implementations."
>
> ### Concern 2: Unsupported Capability Descriptions
> We have revised Section 4.5 (Microsoft Copilot Researcher and Analyst) to properly attribute these claims to their source.
>
> **Revision (Section 4.5):**
> * **Original:** Made direct claims about capabilities without attribution.
> * **Revised:** "These agents securely and compliantly access users’ work data (such as emails, meeting notes, documents, and chats) as well as web information, delivering on-demand expert knowledge. Researcher is designed to assist users in tackling complex, multi-step research tasks, delivering insights with unprecedented quality and accuracy. It combines OpenAI’s advanced research models with Microsoft 365 Copilot’s sophisticated orchestration and deep search capabilities. Users can employ Researcher to craft detailed market entry strategies, identify market opportunities for new products by integrating internal and external data, or prepare comprehensive quarterly reports for client reviews."
>
> ### Concern 3: Marketing Language Masquerading as Evidence
> We have systematically revised the manuscript to adopt a neutral academic tone. Below are specific examples:
>
> **Section 2.1:**
> * **Original:** "notable improvements in both the interpretability and accuracy of LLMs on various reasoning benchmark."
> * **Revised:** "This has led to notable improvements in both the interpretability and accuracy of LLMs on various reasoning benchmarks." (Added specific benchmark references: GSM8K and MATH citations, language neutralized).
>
> **Section 3.4.2:**
> * **Original:** "fundamentally reconfigures the advantage estimation paradigm."
> * **Revised:** "GRPO fundamentally reconfigures the advantage estimation paradigm by replacing traditional value functions with group-relative advantage computation. It expands reward space through intra-group normalisation, and sparse binary rewards are transformed into continuous advantage values spanning wider ranges."
>
> **Section 4.2:**
> * **Original:** "significantly outperforms other agents in terms of retrieval speed."
> * **Revised:** "Implements fast, multi-round adaptive web search that significantly outperforms other agents in terms of retrieval speed and amount of information per iteration."
>
> ### Concern 4: Unclear Evidence in Visualizations
> We have substantially revised all visualizations:
> * **Figure 1:** Added clearer caption: "A structural overview of a DR agent in a multi-agent architecture for ease of illustration."
> * **Table 1:** Changed legend to be more explicit: "■ = Primary focus, ▨ = Secondary/minor focus, □ = Not present"
> * **Table 2:** Reorganized with clearer groupings and updated legend.
>
> ### Concern 5: Future Capabilities Without Basis
> We have grounded this claim by connecting it to the evidence presented in our review.
>
> **Revision (Section 4):**
> > "Looking forward, continuous advancements in LLM reasoning, retrieval integration techniques, and multimodal generation are expected to enable DR agents to transcend traditional information retrieval and basic tool invocation tasks. Consequently, DR systems are anticipated to tackle increasingly sophisticated reasoning and complex knowledge-construction challenges, ultimately positioning DR as a foundational technological pillar for next-generation intelligent collaborative research platforms."
>
> This statement now follows a comprehensive review of industrial applications and is positioned as a forward-looking conclusion based on the documented trajectory of development shown in Figure 5.

---

### Decision · Action_Editor_f8Ev · 2026-03-04

**Recommendation:** Reject

**Additional Comments:**

We receive diverge reviews and final recommendations, where  reviewer zqLR recommends to accept this paper and the other reviewers lean to reject this paper. It is a difficult decision. After carefully considering the reviews and the authors' responses following revision, I recommend rejection of this submission. And below are my justifications of this decision.

The most critical unresolved issue is the internal consistency and analytical depth. The revised manuscript still contains discrepancies between the claims made in the text and what the tables and figures actually show. The most concrete example is in Section 3.4.2, where the text asserts five distinct RL implementation patterns but only three are described.

More fundamentally, the paper categorizes approaches without explaining why the categories matter, what trade-offs exist, or when some approach is preferred than others. A survey paper's primary contribution lies not in listing all existing work but in organizing them and analyzing them to provide an overview to the whole field. The categories proposed should carry explanatory and perspective value for the audience, and this paper can be further improved.

Though rejecting this paper, I do encourage the authors to resubmit the paper after a major revision, as I believe this is an important and relevant topic for this community.

**Audience:**

Yes

**Audience Explanation:**

All reviewers agree about this and so do I because Deep Research Agent is an important and relevant topic for the TMLR community.

**Claims And Evidence:**

No

**Claims Explanation:**

2 of 3 reviewers believe that the claims in the submission are supported with clear evidence and 1 reviewer's answer is no. I think this paper can be further improved. Please see my additional comments for details.

**Resubmission Of Major Revision:**

The authors may consider submitting a major revision at a later time.